# Linear Ensembles Wash Away Watermarks: On the Fragility of Distributional Perturbations in LLMs

**Zhihao Wu** [* 1]  **Gracia Gong** [* 2]  **Qinglin Zhu** [1]  **Yudong Chen** [3]  **Runcong Zhao** [1]

## Abstract

Watermarking embeds statistical signatures in AI-generated text for detection and attribution. We reveal a fundamental vulnerability: when users access multiple models (today's reality), watermarks trivially fail. Watermarks perturb output distributions away from the original, and in competitive markets, these perturbations are typically independent across providers. We theoretically prove that averaging output probability distributions recovers the unwatermarked distribution with up to a second-order error term. Empirically, simply averaging 3-5 models cancels out these perturbations. We introduce WASH (Watermark Attenuation via Statistical Hybridisation), which solves practical challenges in ensemble generation: vocabulary misalignment and tokenisation differences across heterogeneous models. Experiments across six watermarking schemes and three LLMs show that averaging across 3 models suppresses detection z-scores from 5-300 to **below 2** (below the detection threshold of 4) and reduces TPR@5%FPR to **below 50%**, while improving quality by **27.5%** and running **6×** faster than the best baseline on the long sequence generation. Our results suggest that robust AI-text detection via watermarking requires either accepting this fundamental vulnerability or unprecedented coordination among model providers.

## 1. Introduction

The rapid deployment of large language models (LLMs) across critical applications, from educational assessment to content creation, has made reliable attribution mechanisms an urgent necessity (Wang et al., 2024; Khasentino et al., 2025; Wang et al., 2026). Can we determine whether a given text was generated by an AI system? In an era where synthetic text has become increasingly indistinguishable from human writing, these questions have profound implications for academic integrity, content authenticity, and intellectual property protection (Wei et al., 2023; Yao et al., 2024). Watermarking has emerged as the technical solution to this attribution problem (Kirchenbauer et al., 2023a; Liu et al., 2024). By embedding statistical signatures during text generation, unbiased watermarking promises to make AI-generated content detectable while minimising quality degradation and avoiding architectural changes (Hu et al., 2024; Mao et al., 2024).

However, current watermarking research relies on a critical simplifying assumption: adversaries have access to only a single watermarked model. In reality, users today can easily and freely access 10+ frontier LLMs through unified platforms (e.g., Hugging Face), such as GPT, LLaMA, Qwen, Mistral, and dozens of other capable models. The modern LLM landscape is not a monopoly but a hyper-competitive marketplace with multiple providers, and this competitive structure is the Achilles' heel of watermarking. Our key insight is deceptively simple: watermarking works by perturbing a model's output distribution, and these perturbations are independent across providers due to different secret keys and architectural design. By querying multiple models in parallel and averaging their output distributions, these independent perturbations cancel out, recovering the original unwatermarked distribution (as illustrated in Figure 1). We formalise this by proving that for any unbiased watermarking scheme with independent per-model perturbations, linear ensembling recovers the consensus distribution up to a second-order error with convergence rate $O(1/\sqrt{N})$. This establishes a fundamental limit rooted in market structure: competitive providers must use secret keys for provenance verification (guaranteeing independence) and maintain quality to retain users (bounding perturbation magnitude). Under these constraints, watermark signals are mathematically guaranteed to vanish under averaging.

While theoretical result guarantees asymptotic removal, practical deployment faces critical obstacles in efficiency

---

[*]Equal contribution  [1]Department of Informatics, King's College London, UK [2]Department of Mathematics, Imperial College London, UK [3]Department of Statistics, University of Warwick, UK. Correspondence to: Yudong Chen <yudong.chen@warwick.ac.uk>, Runcong Zhao <runcong.zhao@kcl.ac.uk>.

*Proceedings of the 43rd International Conference on Machine Learning*, Seoul, South Korea. PMLR 306, 2026. Copyright 2026 by the author(s).

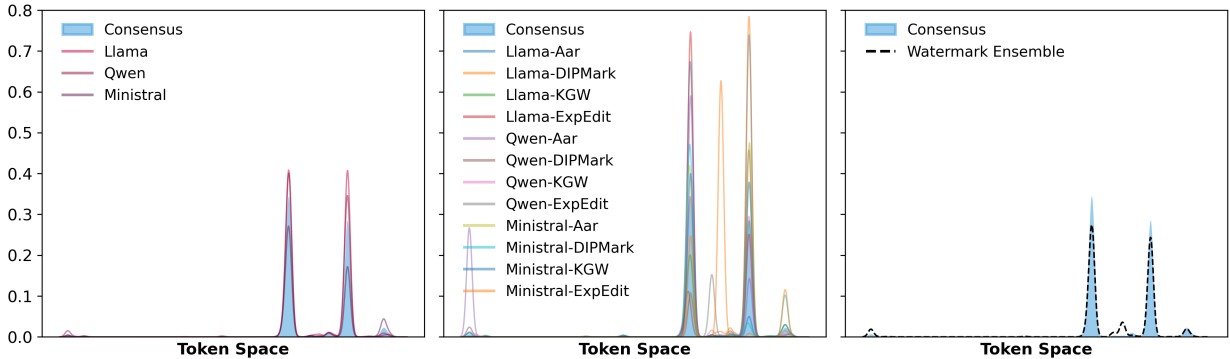

*Figure 1.* **Effect of Linear Ensembles.** The ensemble average $\bar{p}_N$ **(Right)** neutralises the independent watermark perturbations $p_i$ **(Middle)**, effectively recovering the original consensus $p^*$**(Left)**, which is calculated by averaging the unwatermarked models.

and coherence. Frameworks that seek to approximate watermark parameters via "random selection probing" (Chen et al., 2025a) require extensive iterative querying, causing prohibitive inference latency. Meanwhile, attacks that concatenate tokens from heterogeneous models require predicting additional tokens at every time step and re-encoding the entire context at every switch (Huang et al., 2024), incurring significant computational inefficiency. To bridge these gaps, we introduce WASH (Watermark Attenuation via Statistical Hybridisation). WASH employs fluency-aware routing to overcome vocabulary mismatches, enabling effective probability aggregation across distinct tokenisers. Furthermore, by leveraging parallel inference with response caching, WASH eliminates the need for iterative probing or context re-encoding, achieving *constant-time per-token complexity* regardless of the ensemble size $N$. Extensive experiments demonstrate that WASH improves generation quality by $27.5\%$ while running $6\times$ faster than state-of-the-art removal baselines.

Our analysis reveals that watermarking faces a fundamental choice: True robustness that distinguishes AI from human text regardless of model mixing requires **coordinated watermarking**, namely some common signal shared across all models. This paper rigorously characterises this fundamental limitation through the following contributions:

1. We theoretically prove that linear ensembling asymptotically recovers the original unwatermarked distribution. We demonstrate the convergence rate is $O(1/\sqrt{N})$ with $N$ independent models, revealing fundamental limits of unbiased watermarking in multi-provider settings.

2. We introduce WASH, a novel algorithm that overcomes the vocabulary mismatch problem in heterogeneous model ensembles. By employing fluency-aware routing and context re-synchronisation, WASH preserves semantic integrity while neutralising watermark

signals.

3. We conduct a systematic evaluation across six representative watermarking schemes and three LLMs under two complementary detection settings. Experiments demonstrate near-complete detection failure: (a) for generation-time detection, WASH suppresses z-scores from 5-300 (strongly detectable) to $< 2$ (near random choice) with just 3 models; (b) for final-text detection, WASH lowers the TPR@5%FPR on native sequence detectors to below 50%, rendering the watermarks statistically undetectable.

## 2. Methodology

Watermarking exhibits a structural statistical vulnerability when outputs from multiple independently watermarked models are aggregated. Formally, we conceptualise watermarking as a stochastic perturbation applied to a shared underlying distribution $p^*$. While these perturbations are necessary for detection, they act as uncoordinated noise across different providers. Consequently, when outputs from multiple models are combined, the watermark signals interfere destructively, allowing the underlying consensus distribution to be asymptotically recovered.

We now formalise this perspective by characterising the relationship between the consensus distribution, the perturbed watermarked distributions, and their aggregation. Let $\mathcal{V}$ be the vocabulary and $\mathcal{X}$ be the space of possible contexts (i.e., sequences of tokens). We denote the probability distributions over the vocabulary as $\Delta(\mathcal{V}) = \left\{ \mathbf{p} \in \mathbb{R}^{|\mathcal{V}|} \mid p_v \geq 0, \sum_{v \in \mathcal{V}} p_v = 1 \right\}$.

### 2.1. Problem Formulation

Conceptually, the probability distribution of any model $i$ deviates from the ideal semantic distribution $p_{GT}$ (align with human expert) due to shared systematic errors and

provider-specific variations. We model this as:

$$p_i(v|x) \propto p_{GT}(v|x) \cdot \exp\left(\underbrace{\delta_{sys}(v,x)}_{\text{Shared Model Bias}} + \underbrace{\delta_i(v,x)}_{\text{Watermark}}\right).$$

Here, $\delta_{sys}$ represents common biases inherent to current LLM architectures, while $\delta_i$ encapsulates the provider-specific signal, primarily the watermarking signal, but also including model-specific characteristics.

**Definition 2.1** (Consensus Distribution and Watermarked Perturbation). We define the consensus distribution $p^*(\cdot|x) \in \Delta(\mathcal{V})$ as the effective baseline of current models, absorbing the shared systematic bias: $p^*(v|x) \propto p_{GT}(v|x) \cdot \exp(\delta_{sys}(v,x))$. Consequently, the output distribution of a specific watermarked model $i$ defines a perturbed distribution $p_i(\cdot|x)$:

$$p_i(v|x) = \frac{p^*(v|x) \cdot \exp(\delta_i(v,x))}{Z_i(x)}, \quad (1)$$

where $\delta_i : \mathcal{V} \times \mathcal{X} \to \mathbb{R}$ is the provider-specific perturbation function (the watermark signal), and $Z_i(x)$ is the normalisation term.

The key insight enabling watermark removal is that perturbations across independent providers are statistically unbiased. We formalise this assumption below.

**Assumption 2.2** (Unbiased Perturbations). Consider a discrete set of providers indexed by $i$, where each provider is associated with a random perturbation vector $\delta_i(\cdot, x)$ that modulates the output distribution. We assume the following properties hold for $\{\delta_i\}_{i=1}^N$:

**(a) Bounded Magnitude:** The perturbation magnitude is uniformly bounded by a constant $\xi \leq 1$. Specifically, $\|\delta_i(\cdot, x)\|_\infty \leq \xi$ for all $x \in \mathcal{X}$ and providers $i$.

**(b) Independence:** The perturbation vectors $\delta_i$ are independent across providers.

**(c) Zero Mean:** The watermarking signals are centered around the consensus distribution in log-space, such that $\mathbb{E}[\delta_i(v,x)] = 0$ for all $v \in \mathcal{V}$, $x \in \mathcal{X}$, and providers $i$.

**(d) Bounded Expected Variance:** The expected value of the weighted variation of the perturbations across the vocabulary (weighted by the consensus probability) is bounded by a constant $\eta^2$. That is, for all $x \in \mathcal{X}$:

$$\mathbb{E}\left[\text{Var}_{u \sim p^*}(\delta_i(u,x))\right] \leq \eta^2, \quad (2)$$

where

$$\text{Var}_{u \sim p^*}(\delta_i(u,x)) := \sum_u p^*(u)\left(\delta_i(u) - \sum_v p^*(v)\delta_i(v)\right)^2. \quad (3)$$

This assumption is natural in the multi-provider setting: (a) Market forces impose a strict upper bound on perturbation magnitude ($\xi$), as any watermark strong enough to violate the linear approximation (large $\delta$) would result in perceptible quality degradation. (b) Each provider employs watermark configurations unknown to each other, making the perturbations behave as independent random variables. This can be relaxed to a grouped setting where providers share common latent factors: under between-group independence and within-group conditional independence, the same convergence guarantee holds; see Appendix B. (c) Providers independently optimise for quality, with no reason to systematically favor or disfavor specific tokens beyond what $p^*$ suggests. We further stress-test this zero-mean assumption under deliberately biased watermark perturbations in Appendix C. (d) Finally, providers maximise utility by prioritising stability on high-probability tokens; the $p^*$-weighting ensures that significant variance is restricted to rare tokens where it least impacts the overall text quality.

### 2.2. Watermark Removal via Linear Ensembles

Given $N$ independent models, we propose to neutralise the watermarking signals and recover the consensus distribution via a uniform mixture, a method we term WASH. Since the perturbations $\delta_i$ are uncorrelated across providers, they act as noise that can be averaged out. Given a fixed content $x \in \mathcal{X}$, the aggregated probability distribution is defined as

$$\bar{p}_N(\cdot|x) := \frac{1}{N}\sum_{i=1}^N p_i(\cdot|x). \quad (4)$$

We implement this removal process as an autoregressive ensemble. The generation process can be formalised as a recursive sequence. Let $x_{<t} = (x_1, \ldots, x_{t-1})$ denote the context at step $t$. The next token $x_t$ is sampled from the aggregated distribution:

$$x_t \sim \bar{p}_N(\cdot|x_{<t}).$$

The process repeats until a termination token is generated. Having formalised the generation process, we now provide theoretical guarantees for its effectiveness.

**Theorem 2.3** (Convergence to Consensus Distribution). *Under Assumption 2.2, for any fixed context $x$, let $\bar{p}_N(\cdot|x) = \frac{1}{N}\sum_{i=1}^N p_i(\cdot|x)$ be the aggregated distribution. For any $\delta > 0$, with probability at least $1 - \delta$, the $\ell_\infty$ distance between the aggregated distribution and the consensus distribution $p^*(\cdot|x)$ satisfies:*

$$\left\|\bar{p}_N(\cdot|x) - p^*(\cdot|x)\right\|_\infty \lesssim \sqrt{\frac{\log(|\mathcal{V}|/\delta)}{N}} + \eta^2, \quad (5)$$

*where $|\mathcal{V}|$ denotes the vocabulary size, and $\eta^2$ is the upper bound on the expected weighted variance of the perturbation as in (2).*

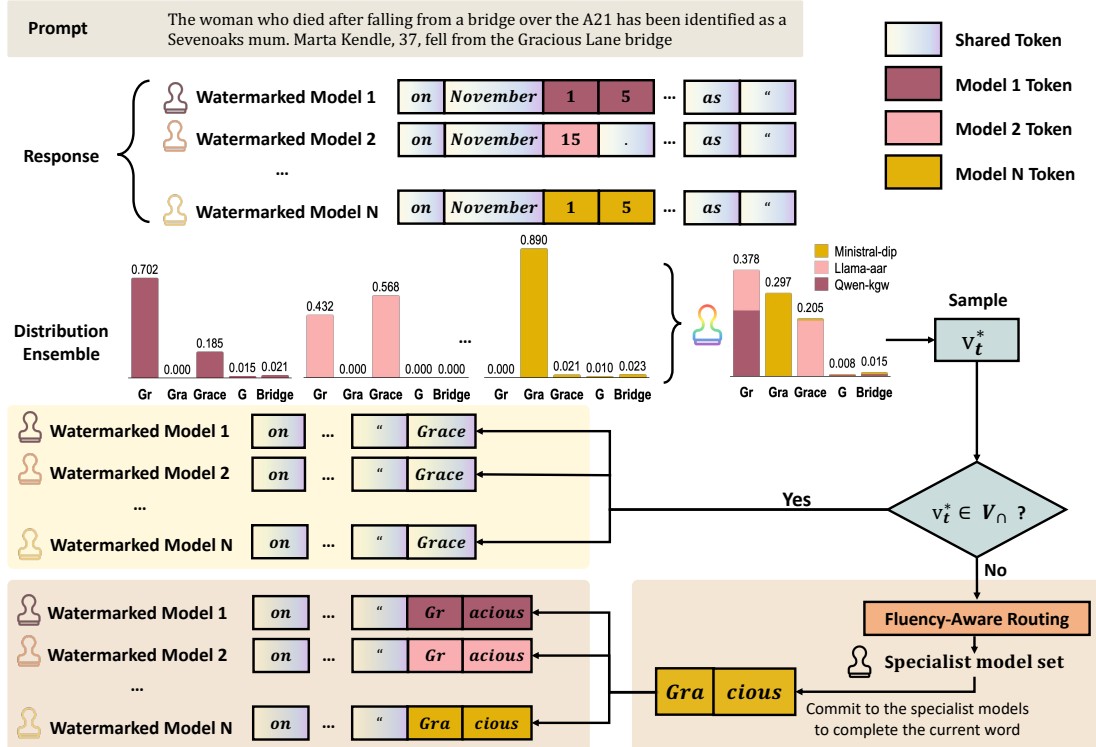

*Figure 2.* **WASH Framework Overview.** The method ensembles $N$ independent models to neutralise watermarks via probability averaging. To resolve vocabulary mismatches, *Fluency-Aware Routing* commits to the specialist models ($K^*$) whenever a token falls outside the shared vocabulary intersection ($\mathcal{V}_\cap$), employing *re-tokenisation* to synchronise context and ensure semantic alignment across heterogeneous tokenisers.

*Proof Sketch.* The proof relies on decomposing the approximation error into a *stochastic deviation term* and a *systematic bias term*. By the triangle inequality, the distance between the aggregated distribution $\bar{p}_N$ and the consensus distribution $p^*$ can be bounded as:

$$\|\bar{p}_N - p^*\|_\infty \leq \underbrace{\|\bar{p}_N - \mathbb{E}[\bar{p}_N]\|_\infty}_{\text{Stochastic Deviation}} + \underbrace{\|\mathbb{E}[\bar{p}_N] - p^*\|_\infty}_{\text{Systematic Bias}}.$$

*Bounding the Stochastic Deviation.* Since the providers are independent (Assumption 2.2(b)), the aggregated probability $\bar{p}_N(v)$ is the average of independent bounded random variables. A direct application of Hoeffding's inequality shows that $\bar{p}_N$ concentrates around its expectation $\mathbb{E}[p_i]$ at a rate of $O(1/\sqrt{N})$.

*Bounding the Systematic Bias.* The term $\|\mathbb{E}[p_i] - p^*\|_\infty$ arises from the nonlinearity of the softmax function. Although the perturbations $\delta_i$ are zero-mean (Assumption 2.2(c)), the expected output probability is biased (i.e., $\mathbb{E}[p_i] \neq p^*$) due to the convexity of the exponential function. We utilise the shift-invariance of the softmax to centre the perturbations and apply a second-order Taylor expansion to $p_i(v) \propto p^*(v) \exp(\delta_i(v))$. The first-order (linear) terms vanish due to the zero-mean assumption. The remaining

error is dominated by the second-order terms, which are controlled by the variance bound $\eta^2$ defined in Assumption 2.2(d). We provide the detailed proof in Appendix A. □

### 2.3. Preserving Semantic Integrity in Heterogeneous Ensembles

While linear ensemble effectively neutralises watermark perturbations in the limit of $N$, strictly enforcing this over heterogeneous topologies introduces a vocabulary mismatch problem (Yu et al., 2024; Chen et al., 2025b). Let $\mathcal{V}_i$ denote the vocabulary of model $i$. A standard baseline restricts sampling to the vocabulary intersection $\mathcal{V}_\cap = \bigcap_{i=1}^N \mathcal{V}_i$, thereby ensuring distributional consensus across all models. However, this conservative approach induces *expressivity bottleneck*. Restricting generation to the vocabulary intersection $\mathcal{V}_\cap$ often excludes semantic ground truth tokens, such as specific entities or technical terms that are absent in at least one model's vocabulary (i.e., $\mathcal{V}_\Delta = (\bigcup \mathcal{V}_i) \setminus \mathcal{V}_\cap$).

Conversely, projecting onto the vocabulary union $\mathcal{V}_\cup = \bigcup_{i=1}^N \mathcal{V}_i$ introduces *granularity mismatch*: as illustrated in Figure 2, heterogeneous tokenisers may represent the same semantic unit at different granularities. For instance, one model represents "Gracious" as "[Gr], [acious]" while an-

other decomposes it into "[Gra], [cious]". Aggregating over the union forces the imputation of undefined probabilities for different disjoint tokens (e.g., the token "[Gra]" does not exist in the first model's vocabulary), thereby diluting the semantic fidelity. To reconcile vocabulary heterogeneity with watermark removal, we propose a dynamic switching process that routes generation between the ensembled distribution and a set of local "specialist" models based on support constraints. The generation at step $t$ is determined by the policy outlined below:

**Generation via Union-based Ensemble**   To avoid semantic truncation, we construct the ensemble distribution $\bar{p}_N$ over the vocabulary union $\mathcal{V}_\cup$. For any model $i$ with token $v \notin \mathcal{V}_i$, we assign $p_i(v|x_{<t}) = 0$, so that the ensemble average reduces to $\bar{p}_N(v|x_{<t}) = \frac{1}{N} \sum_{i:v \in \mathcal{V}_i} p_i(v|x_{<t})$. For any token supported by only one model, this caps $\bar{p}_N(v|x_{<t})$ at approximately $1/N$ of that model's own probability. However, this scaling does not indicate a loss of confidence: models lacking $v$ in their vocabulary still allocate comparable probability mass to the same semantic content, but route it through alternative tokenisation paths (e.g., one model emits "Gracious" as an atomic token while others emit the same word as "[Gr][acious]"). The semantic signal is therefore dispersed across divergent tokenisation paths rather than discarded. Crucially, unlike the intersection approach, every token in $\mathcal{V}_\cup$ remains accessible. The next token is then sampled from this union-aggregated distribution: $x_t \sim \bar{p}_N(\cdot|x_{<t})$.

**Fluency-Aware Routing**   If $x_t \in \mathcal{V}_\cap$ (i.e., the token is valid across all models), we directly output $x_t$. However, a critical challenge arises when the selected token $x_t \in \mathcal{V}_\Delta$, it is undefined to the subset of models that lack this token in their vocabulary. Accepting such a token would break the autoregressive chain, as the incompatible models cannot process $x_t$ as valid input context for the subsequent generation step $(t + 1)$. To preserve fluency, we employ a randomised routing mechanism that restricts subsequent sampling to "specialist" models whose vocabularies admit the committed tokens. Let $K$ denote the set of all available models and we initialise $K_t = K$. For each $t' \in [t+1, \tau(t)]$, where $\tau(t)$ denotes word completion, we recurrently update the specialist set as

$$K_{t'} = \{i : x_{t'-1} \in \mathcal{V}_i\} \cap K_{t'-1}.$$

We then uniformly sample a model from the specialist set before generating the token at $t'$:

$$k_{t'} \sim \text{Uniform}(K_{t'}), \quad x_{t'} \sim p_{k_{t'}}(\cdot|x_{<t'}).$$

Crucially, we avoid selecting the model using maximum likelihood principles, as likelihood disparities may themselves be watermark-induced artifacts. Random selection

ensures the routing decision is orthogonal to watermark signals, preventing detectors from exploiting systematic biases in model selection.

While routing temporarily re-admits the watermark signal $\delta_{\text{wm}}^{(k_{t'})}$, such events are sparse and confined to vocabulary boundaries. Moreover, our strategy consistently engages as many specialist models as possible, ensuring that ensembling remains active throughout the generation. Critically, even if some watermark signal exhibits, watermark artefacts are temporally fragmented: they alternate stochastically across different specialists and are interspersed with non-routing tokens. This prevents the accumulation of sustained statistical regularities required for reliable watermark identification. We provide a complete generation example in Appendix E.

**Context Re-synchronisation**   Routing to specialist models creates **context de-synchronisation**: non-selected models do not observe the generated tokens in their native tokenisation. To maintain coherence, we apply a decode-encode cycle. We decode the generated string with the last selected specialist model $S_t = \text{Decode}_{k_{\tau(t)}}(\hat{x}_{t:\tau(t)})$, and update each model's context via:

$$x^{(i)}_{<\tau(t)+1} \leftarrow x^{(i)}_{<t} \oplus \mathcal{T}_i(S_t), \quad \forall i \in [N], \qquad (6)$$

where $\mathcal{T}_i$ is model $i$'s tokeniser. This ensures all models observe semantically equivalent contexts despite tokenisation differences, preserving valid probability aggregation in subsequent ensemble steps.

## 3. Experiment

### 3.1. Experimental Setup

**Models and Watermarks.**   We conduct experiments on three widely used pre-trained LLMs: `Qwen3-8B` (Qwen, 2025), `Llama-3.1-8B` (Meta AI, 2024), and `Ministral3-8B` (Mistral, 2026). To ensure broad coverage, we evaluate six representative watermarking schemes spanning different design paradigms: `AAR`, which relies on uniform distribution (Aaronson & Kirchner, 2022); `DIPMark`, which applies logit reweighting (Wu et al., 2024); `ITSEdit`, based on inverse transform sampling with exponential minimum sampling (Kuditipudi et al., 2024); `KGW`, which encourages sampling to a green list of tokens (Kirchenbauer et al., 2023a); `Exp-Edit`, which uses key-based transformations (Kuditipudi et al., 2024); and `Water-Bag`, which combines a set of watermark keys and their mathematical inversions during generation to mask statistical biases (Liu et al., 2025).

**Baselines.** We compare WASH with two classes of watermark removal attacks.

*Generation-time attacks* produce the output directly under

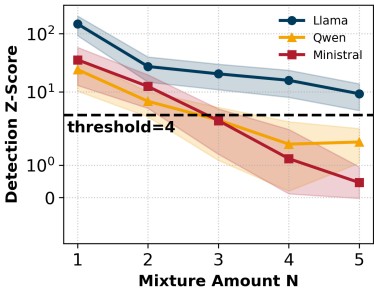 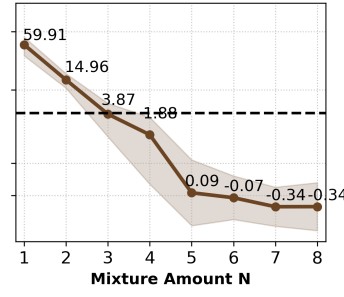 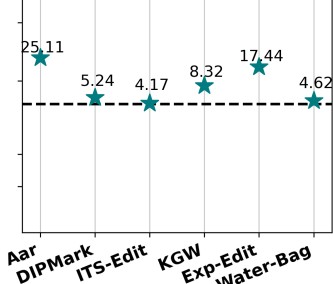

*Figure 3.* **Watermark signal decay under different ensemble configurations.** We measure detection strength (z-score) as ensemble size $N$ increases. **(a)** Fixed base model with $N$ independent watermark keys. **(b)** $N$ independent models, each with independent watermarks. **(c)** 3 independent base models sharing the same watermark, with the signals coordinated across models: signal persists, demonstrating that coordination defeats averaging attacks.

the removal procedure, matching the WASH setting:
(1) `De-mark` (Chen et al., 2025a): A watermark removal method designed against red-green-list watermarking. It first identifies the green-list tokens whose logits are biased by the watermark using crafted prompts, then removes the bias to restore the original generation distribution.
(2) `ToBlend` (Huang et al., 2024): A model generation mixture method against AI text detections. It blends multiple model generations by predicting a fixed number of tokens with a single selected model each time.

*Final-text rewrite attacks* instead operate on an already generated watermarked sequence:
(1) `RandomWalk` (Liu et al., 2025): A strategy that repeatedly rewrites spans of the generated text using a weaker unwatermarked model and accepts only quality-preserving variants based on a quality-check oracle.

**Benchmarks and Metrics.** We use two evaluation suites:
(1) **Detection**. For *generation-time attacks*, we follow the z-score protocol of Liu et al. (2025) to quantify watermark signal strength. For *final-text rewrite attacks*, perturbation detection on the small set of generated tokens is no longer compatible, so we additionally use native sequence detectors (Pan et al., 2024). Detection thresholds and confidence categories are reported in Appendix D.1. (2) **Quality and Efficiency**. To assess generation quality and inference speed, we use four benchmarks covering representative reasoning and language tasks: GSM8K (math reasoning) (Cobbe et al., 2021), MMLU (knowledge) (Hendrycks et al., 2021), SQuAD (reading comprehension) (Rajpurkar et al., 2016), WritingBench (open-ended writing) (Wu et al., 2025).

### 3.2. Watermark Removal Effectiveness

The experimental results in Figure 3 demonstrate the inherent fragility of distributional watermarking when subjected to linear ensemble averaging. Our analysis proceeds in three stages: validating the scaling law of signal decay, isolating

the removal mechanism through control experiments, and comparing comprehensive performance against baselines.

**Signal Decay: From Theory to Reality.** We first validate the scaling law of signal decay in an idealised setting where models utilise independent watermarks on the same base model (Figure 3(a)). The detection signal diminishes rapidly with $N$, mirroring our theoretical prediction of $O(1/\sqrt{N})$. The pivotal result lies in Figure 3(b), which simulates the realistic user scenario: accessing distinct heterogeneous models (e.g., Llama, Qwen, Ministral) with independent watermarks. Despite the challenge of vocabulary mismatches, our fluency-aware routing successfully neutralises the watermark signal. Notably, the detection strength drops even faster than in the homogeneous setting (reaching near $z = 0$ at $N = 5$), suggesting that the diversity of base model distributions acts as additional noise that further obscures the watermark trace. We provide the experiment details in Appendix D.2.

**A Control Experiment: Coordinated Watermark.** To verify the mechanism of removal, we evaluate a "Coordinated Watermark" scenario where all ensemble models share the same watermark scheme, and signals are coordinated across distinct vocabularies during each token generation. Specifically, we calculate the relative perturbations of the common tokens from one watermarked model and map them to other models using the same relative scale. As illustrated in Figure 3(c), when the ensemble models are synchronised, the washing effect fails. The ensemble average retains a statistically significant z-score well above the detection threshold ($z > 4$) across all schemes. Strong schemes like AAR and Exp-Edit maintain high z-scores of 25.11 and 17.44, respectively. Crucially, the *persistence* of these signals demonstrates that the ensemble process itself does not inherently obliterate watermark information. Instead, the removal efficacy in the independent setting (Figure 3(a,b)) exploits the lack of correlation between providers. This finding highlights a critical defensive insight: robust AI-text detection

*Table 1.* Comparison of Watermark Removal Effectiveness. The table evaluates z-score after various removal attacks. �In indicates high-confidence watermark identification, and ▢ indicates low-confidence watermark identification, while no colour indicates no watermark identified.

| Base Model | Removal Method | Target Watermark (Individual) | | | | | |
|---|---|---|---|---|---|---|---|
| | | AAR | DIPMark | ITS-Edit | KGW | Exp-Edit | Water-Bag |
| Llama3.1-8B | Watermarked Baseline | 176.438 | 58.127 | 304.022 | 50.113 | 61.981 | 10.237 |
| | De-mark | — | 99.281 | — | 1.786 | — | -10.687 |
| | ToBlend | 3.636 | 1.344 | 0.489 | 0.819 | 0.482 | -0.003 |
| | **WASH (N=3)** | 3.699 | 1.477 | 0.384 | 0.634 | -0.245 | -0.109 |
| | **WASH (N=4)** | 3.082 | 0.563 | 0.408 | 0.550 | 0.372 | 0.093 |
| | **WASH (N=5)** | 2.232 | 0.182 | 0.119 | 0.372 | -0.108 | -0.111 |
| Qwen3-8B | Watermarked Baseline | 15.534 | 14.210 | 15.418 | 34.547 | 26.904 | 5.245 |
| | De-mark | — | 4.560 | — | 3.770 | — | -8.452 |
| | ToBlend | 1.941 | 0.523 | 0.464 | 0.489 | 0.175 | -0.169 |
| | **WASH (N=3)** | 1.629 | 0.236 | 0.602 | 0.058 | 0.517 | -0.205 |
| | **WASH (N=4)** | 1.286 | 0.094 | 0.068 | 0.130 | 0.200 | 0.000 |
| | **WASH (N=5)** | 0.654 | 0.189 | 0.337 | 0.222 | 0.079 | -2.570 |
| Ministral3-8B | Watermarked Baseline | 9.175 | 23.135 | 27.082 | 27.407 | 59.507 | 9.568 |
| | De-mark | — | 91.174 | — | -2.814 | — | 0.729 |
| | ToBlend | 1.204 | 1.556 | 0.711 | 1.168 | 0.623 | 0.437 |
| | **WASH (N=3)** | 1.070 | 0.059 | 0.558 | 1.732 | 0.787 | 0.775 |
| | **WASH (N=4)** | 1.901 | 1.350 | 1.363 | 1.602 | 0.630 | -1.809 |
| | **WASH (N=5)** | 0.421 | 0.141 | 0.456 | 0.461 | 0.435 | -0.382 |

against ensemble attacks requires *coordinated watermarking strategies* across model providers.

**Comparison with Generation-time Attacks.** Table 1 compares WASH against *generation-time attacks* across various LLMs using the same z-score detection protocol. Individual watermarked models exhibit strong detection signals, with z-scores ranging from 5.2 to 304.0. Prior removal methods show limited efficacy: `De-mark` performs well only on pure red-green-list watermarking methods with exact logits bias value like KGW and Water-Bag, while collapsing on other distribution modifications (4.6 to 99.3 z-score on DIP-Mark). It also cannot apply to watermarks that operate on probabilities. In contrast, WASH consistently neutralises detection signals. An ensemble of $N = 3$ proves practically sufficient to suppress z-scores below the threshold ($z < 4$) across all cases. Extending to $N = 5$ offers a safety margin for aggressive schemes, ensuring near-zero distinguishability. `ToBlend`, as a model mixture method like ours, could achieve similar removal efficacy to WASH in the detection evaluation, while significantly harming generation quality and efficiency, as we discuss in the next section.

*Table 2.* Native-detector watermark removal results with final-text rewrite attacks. The results are reported in TPR@5% FPR calibrated on unwatermarked results, where lower is better.

| Method | DIPMark | KGW | AAR | ITS-Edit | Exp-Edit |
|---|---|---|---|---|---|
| Watermarked | 83.7 | 92.8 | 95.9 | 83.5 | 85.9 |
| De-mark | 76.5 | 58.8 | — | — | — |
| ToBlend | 28.2 | 38.4 | 45.4 | 21.4 | 11.2 |
| RandomWalk | 49.0 | 56.7 | 24.2 | 53.6 | 24.0 |
| **WASH (N=5)** | 27.6 | 38.9 | 42.8 | 16.2 | 11.3 |

**Comparison with Final-text Rewrite Attacks.** *Final-text rewrite attacks* require a separate evaluation as they

perturb the completed sequence, making token perturbation detectors inapplicable. We thus report the TPR@5% FPR from native sequence-level detectors. Lower values indicate stronger removal under thresholds calibrated on unwatermarked samples. As shown in Table 2, `RandomWalk` leaves several schemes near or within the low-confidence range. WASH keeps all schemes below 43%, matching `ToBlend`'s removal ability and outperforming `RandomWalk` on 4 out of 5 schemes, while retaining quality and efficiency advantages analysed below.

### 3.3. Quality, Semantic Integrity, and Efficiency

**Comparison with Generation-time Attacks.** As shown in Table 3, watermarking inevitably distorts the optimal output distribution, leading to up to 10% performance degradation relative to the unwatermarked baseline. While removal attacks aim to recover this lost utility by restoring the original distribution, prior methods face two bottlenecks: (1) excessive perturbation may further harm generation quality, and (2) the computational cost of recovery can be prohibitive.

Consistent with our theoretical analysis, WASH resolves these tensions by recovering the unwatermarked distribution through ensembling. It achieves generation quality comparable to or superior to the strongest baseline (`De-mark`) in 7 out of 9 benchmark settings, while outperforming the other mixture method (`ToBlend`) across all settings.

The efficiency advantage is decisive for generation-intensive tasks. While prior overhead may be tolerable for short discriminative tasks such as MMLU, it becomes prohibitive for long-form generation, where watermarking is most relevant for content provenance, copyright protection, and misuse

*Table 3.* **Performance and Efficiency Comparison.** We evaluate Blue generation quality (GSM8K, MMLU, SQuAD) and Red relative computational cost (Time) across the watermarked baseline, removal attacks, and our proposed method with varying ensemble sizes ($n$). Time is normalised to the baseline cost ($1.0\times$).

| Base Model | Method | GSM8K | | MMLU | | SQuAD | |
|---|---|---|---|---|---|---|---|
| | | Acc. ↑ | Time ↓ | Acc. ↑ | Time ↓ | F1 ↑ | Time ↓ |
| Llama3.1-8B | Unwatermarked | $0.567 \pm 0.012$ | $1.00\times$ | $0.583 \pm 0.000$ | $1.00\times$ | $0.795 \pm 0.013$ | $1.00\times$ |
| | Watermarked Baseline | $0.511 \pm 0.032$ | $1.00\times$ | $0.560 \pm 0.023$ | $1.00\times$ | $0.749 \pm 0.038$ | $1.00\times$ |
| | De-mark | $0.550 \pm 0.040$ | $38.80\times$ | $0.565 \pm 0.015$ | $10.01\times$ | $0.642 \pm 0.071$ | $32.50\times$ |
| | ToBlend | $0.568 \pm 0.087$ | $14.60\times$ | $0.631 \pm 0.038$ | $1.06\times$ | $0.633 \pm 0.045$ | $17.55\times$ |
| | **WASH (N=3)** | $0.695 \pm 0.067$ | $2.19\times$ | $0.631 \pm 0.038$ | $1.06\times$ | $0.764 \pm 0.031$ | $1.49\times$ |
| | **WASH (N=4)** | $0.701 \pm 0.074$ | $2.37\times$ | $0.644 \pm 0.032$ | $1.06\times$ | $0.765 \pm 0.028$ | $1.53\times$ |
| | **WASH (N=5)** | $0.698 \pm 0.057$ | $2.28\times$ | $0.646 \pm 0.025$ | $1.06\times$ | $0.764 \pm 0.031$ | $1.54\times$ |
| Qwen3-8B | Unwatermarked | $0.713 \pm 0.021$ | $1.00\times$ | $0.726 \pm 0.000$ | $1.00\times$ | $0.421 \pm 0.004$ | $1.00\times$ |
| | Watermarked Baseline | $0.693 \pm 0.051$ | $1.00\times$ | $0.723 \pm 0.002$ | $1.00\times$ | $0.396 \pm 0.014$ | $1.00\times$ |
| | De-mark | $0.755 \pm 0.035$ | $30.97\times$ | $0.724 \pm 0.002$ | $9.47\times$ | $0.434 \pm 0.012$ | $30.98\times$ |
| | ToBlend | $0.701 \pm 0.050$ | $11.40\times$ | $0.722 \pm 0.011$ | $1.01\times$ | $0.474 \pm 0.071$ | $19.18\times$ |
| | **WASH (N=3)** | $0.804 \pm 0.050$ | $1.53\times$ | $0.722 \pm 0.011$ | $1.01\times$ | $0.610 \pm 0.084$ | $1.69\times$ |
| | **WASH (N=4)** | $0.809 \pm 0.047$ | $1.58\times$ | $0.723 \pm 0.009$ | $1.01\times$ | $0.629 \pm 0.067$ | $1.82\times$ |
| | **WASH (N=5)** | $0.808 \pm 0.035$ | $1.60\times$ | $0.723 \pm 0.010$ | $1.01\times$ | $0.612 \pm 0.071$ | $1.96\times$ |
| Ministral3-8B | Unwatermarked | $0.837 \pm 0.025$ | $1.00\times$ | $0.731 \pm 0.000$ | $1.00\times$ | $0.839 \pm 0.000$ | $1.00\times$ |
| | Watermarked Baseline | $0.809 \pm 0.041$ | $1.00\times$ | $0.716 \pm 0.027$ | $1.00\times$ | $0.795 \pm 0.032$ | $1.00\times$ |
| | De-mark | $0.805 \pm 0.005$ | $36.39\times$ | $0.717 \pm 0.015$ | $10.02\times$ | $0.832 \pm 0.004$ | $36.42\times$ |
| | ToBlend | $0.715 \pm 0.036$ | $12.17\times$ | $0.716 \pm 0.015$ | $1.04\times$ | $0.775 \pm 0.016$ | $15.92\times$ |
| | **WASH (N=3)** | $0.815 \pm 0.030$ | $1.25\times$ | $0.716 \pm 0.015$ | $1.04\times$ | $0.805 \pm 0.026$ | $1.19\times$ |
| | **WASH (N=4)** | $0.821 \pm 0.026$ | $1.30\times$ | $0.718 \pm 0.007$ | $1.03\times$ | $0.807 \pm 0.020$ | $1.30\times$ |
| | **WASH (N=5)** | $0.818 \pm 0.012$ | $1.28\times$ | $0.718 \pm 0.005$ | $1.03\times$ | $0.821 \pm 0.029$ | $1.30\times$ |

attribution. On GSM8K and SQuAD, `De-mark` suffers severe latency ($> 30\times$) due to iterative distribution probing, and `ToBlend` incurs similarly high overhead ($\sim 12\times$) because it repeatedly reprocesses context and predicts redundant future tokens. In contrast, WASH operates at only $1.0\times$–$2.4\times$ the baseline cost through deterministic sampling and synchronised KV caching, giving roughly a $6\times$ speedup over competing methods while maintaining stronger quality.

**Comparison with Final-text Rewrite Attacks.** We evaluate `RandomWalk` on GSM8K and WritingBench rather than MMLU and SQuAD because final-text rewriting requires sufficiently long generations to meaningfully change surface form while preserving content. MMLU multiple-choice outputs and SQuAD short answers are often too short to expose either rewriting effectiveness or its utility cost.

*Table 4.* Comparison with final-text rewriting on GSM8K and WritingBench. Runtime is normalised to the watermarked baseline.

| Method | GSM8K | | WritingBench | |
|---|---|---|---|---|
| | Acc. ↑ | Time ↓ | Score ↑ | Time ↓ |
| Watermarked | 0.511 | $1.00 \times$ | 4.10 | $1.00 \times$ |
| De-mark | 0.550 | $38.46 \times$ | 4.04 | $43.97 \times$ |
| ToBlend | 0.568 | $13.28 \times$ | 2.32 | $7.89 \times$ |
| RandomWalk | 0.467 | $4.39 \times$ | 3.61 | $10.03 \times$ |
| **WASH (N=5)** | **0.698** | $2.28 \times$ | **4.26** | $1.85 \times$ |

As shown in Table 4, `RandomWalk` underperforms the watermarked baseline on both reasoning and open-ended writing, likely because in-place rewriting with another model introduces instability. Its additional rewriting phase also incurs substantial overhead, reaching $4\times$ runtime on GSM8K

and $10\times$ on WritingBench. In contrast, WASH improves quality while requiring only about $2\times$ runtime. Thus, final-text rewriting may weaken detection, but it pays for this through lower utility and significantly higher latency.

**Resource and Serving Trade-offs.** Table 5 further evaluates the system's cost of WASH. The parallel implementation keeps all specialists resident and synchronises their KV caches, prioritising latency. The sequential implementation loads specialists on demand, reducing peak memory at the cost of higher token latency. This gives WASH a practical deployment trade-off: WASH-Par. is substantially faster than `De-mark` and `ToBlend`, while WASH-Seq. reduces peak memory close to the single-model baseline. Thus, WASH remains practical under both latency-constrained and memory-constrained serving regimes.

*Table 5.* Resource usage and decoding latency. Peak memory is measured during generation, and token latency is averaged over generated tokens. WASH-Par. keeps all specialists resident, whereas WASH-Seq. offloads inactive specialists dynamically.

| Method | Peak Mem | Rel. ↓ | Token Latency | Rel. ↓ |
|---|---|---|---|---|
| Unwatermarked | 15.09 GB | $1.00\times$ | 29.5 ms | $0.98\times$ |
| Watermarked | 15.10 GB | $1.00\times$ | 30.0 ms | $1.00\times$ |
| De-mark | 16.58 GB | $1.10\times$ | 190.6 ms | $6.35\times$ |
| ToBlend | 71.46 GB | $4.73\times$ | 172.3 ms | $5.74\times$ |
| **WASH-Par.** | 40.87 GB | $2.71\times$ | **56.9 ms** | **1.90×** |
| **WASH-Seq.** | **15.76 GB** | **1.04×** | 165.5 ms | $5.52\times$ |

### 3.4. Ablation and Analysis of Fluency-Aware Routing

**Routing Ablation.** Fluency-aware routing is the main difference between WASH and naive distribution averaging.

Table 6 evaluates this design choice. `Naive Avg` removes routing and directly averages distributions. The `Rewrite` variants then attempt to repair the `Naive Avg` output using an additional watermarked or unwatermarked copy of the same base model, testing whether post-hoc rewriting can replace generation-time routing.

*Table 6.* Ablation of fluency-aware routing.

| Method | GSM8K ↑ | WritingBench ↑ | Detect TPR@5% FPR ↓ |
|---|---|---|---|
| Naive Avg | 0.339 | 3.95 | 41.3 |
| + Rewrite (WM) | 0.197 | 4.02 | 76.8 |
| + Rewrite (Un-WM) | 0.205 | 4.05 | 10.9 |
| **WASH (N=5)** | **0.698** | **4.26** | 33.3 |

The results show that post-hoc repair is not an effective substitute for routing. `Naive Avg` suppresses detection, but substantially degrades GSM8K accuracy. `Rewrite` improves surface-level writing quality, yet it further hurts reasoning accuracy. WASH instead performs sparse local routing during generation, preserving both low detection and substantially stronger utility.

**Routing Robustness.** We further test whether fluency-aware routing could reintroduce watermark signal when specialised vocabulary is frequent. This concern is most relevant in domains such as medicine and law, where rare terminology might trigger long single-specialist spans, weakening the averaging effect of WASH. Table 7 shows that this does not occur. The routed tokens account for less than 3.2% of generation; routed spans are short; and final detection scores remain far below the watermark threshold. Manual inspection suggests that complex technical terms are often decomposed into shared subword units, while routing is more frequently triggered by ordinary lexical units with tokeniser mismatches. Thus, fluency-aware routing acts as a sparse local repair mechanism rather than sustained single-model generation.

*Table 7.* Routing statistics under specialised vocabulary. Medical and legal MMLU subsets are converted into free-form reasoning prompts. We report routing frequency, routed-span length, and final detection score.

| Domain | Routed Token Frac. | Avg. Routed Len | Detection $z$ ↓ |
|---|---|---|---|
| Medical | 3.2% | 3.7 | 0.84 |
| Legal | 2.7% | 3.6 | 1.08 |

## 4. Related Works

**Providing and Detecting Watermarks**. Recent research has established LLM watermarking through green/red lists. (Kirchenbauer et al., 2023a). While effective, this approach struggles with low-entropy text and is prone to Type II errors. To mitigate these limitations, subsequent research has focused on distribution-preserving and resilient schemes. For instance, Kuditipudi et al. (2024) and Wu et al. (2024) propose distortion-free methods to maintain generation quality. Similarly, Hu et al. (2024) introduce unbiased watermarking via reweighting mechanisms and log-likelihood tests; however, this method assumes knowledge of the generation process and often lacks adversarial robustness. Further advancements explore alternative mechanisms on modifying generation bias through a semantic-aware watermarking (Guo et al., 2024) or maximal coupling strategies (Xie et al., 2025), while others aim to tackle detection challenges in low-entropy scenarios (Mao et al., 2024) or improve scalability (Dathathri et al., 2024).

**Watermark Removal Attacks**. Early attack approaches use paraphrasing to preserve semantics while resampling expressions (Krishna et al., 2023), or sampling from multiple keys to perform majority voting (Pang et al., 2024). RANDOMWALK further rewrites local spans and accepts quality-preserving candidates through an external quality check (Liu et al., 2025). However, these methods typically rely on detector-available scenarios, assuming the adversary has access to an oracle verifier. Recent research attempts to eliminate this dependence. Many theoretical attacks necessitate query-intensive strategies to reverse-engineer watermarking rules. Methods such as WATERMARK STEALING (Jovanović et al., 2024), SCTS (Wu & Chandrasekaran, 2024), and DE-MARK (Chen et al., 2025a) require a high volume of specific prompt queries or iterative token-level probing to infer red-green lists. Similarly, optimisation-based attacks employ computationally expensive random walks (Zhang et al., 2024). While theoretically effective, these strategies are impractical for real-time or long-form generation tasks due to prohibitive latency and computational costs. Additionally, TOBLEND attempts to bypass watermarks by alternating between models (Huang et al., 2024), though often at the cost of coherence or inference efficiency due to frequent context re-encoding.

## 5. Conclusion

We demonstrate that current distributional watermarking schemes are structurally vulnerable to linear ensembling. Theoretically, we prove that averaging outputs from independent models cancels watermark perturbations. To address vocabulary mismatches in heterogeneous ensembles, we introduce WASH, which utilises fluency-aware routing to enable effective probability aggregation. Empirically, WASH renders watermarks statistically undetectable ($z < 2$) with as few as three models, while preserving generation quality with practical inference efficiency. Our findings suggest that reliable detection in a competitive marketplace is unattainable without industry-wide standardisation of watermark keys among model providers.

## Impact Statement

This paper presents work whose goal is to advance the field of AI-generated content detection. This is increasingly important for maintaining societal trust in digital information and content authenticity, and for protecting intellectual property. We demonstrate that various contemporary LLM watermarking schemes are fragile under a general linear ensembling attack. While this attack exposes current limitations, it also serves as an alert to model providers and researchers to strengthen watermarking against adaptive and cross-model threats.

Moreover, we have conducted a coordination experiment, revealing that even simple signal coordination across heterogeneous models can partially mitigate the effectiveness of the ensemble attack. These findings suggest that true watermarking robustness depends not only on isolated model-level defences, but also on cooperative ways that may be more effective. In general, we underscore the value of cross-provider collaboration in developing robust watermarking provenance mechanisms for AI-generated content.

## Acknowledgments

This work was supported in part by the UK Engineering and Physical Sciences Research Council (EPSRC) through a Turing AI Fellowship (grant no. EP/V020579/1, EP/V020579/2). We thank Xiaojia Rao for participating in the initial discussions that helped shape the idea of this work.

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

# A. Proof of the Main Theorem

**Theorem A.1** (Convergence to Consensus Distribution). *Under Assumption 2.2, for any fixed context $x$, let $\bar{p}_N(\cdot|x) = \frac{1}{N}\sum_{i=1}^{N} p_i(\cdot|x)$ be the aggregated distribution. For any $\delta > 0$, with probability at least $1 - \delta$, the $\ell_\infty$ distance between the aggregated distribution and the consensus distribution $p^*(\cdot|x)$ satisfies:*

$$\left\| \bar{p}_N(\cdot|x) - p^*(\cdot|x) \right\|_\infty \lesssim \sqrt{\frac{\log(|\mathcal{V}|/\delta)}{N}} + \eta^2,$$

*where $|\mathcal{V}|$ denotes the vocabulary size, and $\eta^2$ is the upper bound on the expected weighted variance of the perturbation as in (2).*

*Proof.* For brevity, we omit the dependency on $x$ in the notation throughout this proof (e.g., $p^*(v)$ instead of $p^*(v|x)$). The perturbed distribution for model $i$ is given by:

$$p_i(v) = \frac{p^*(v)\exp(\delta_i(v))}{Z_i}, \quad \text{where } Z_i = \sum_{u \in \mathcal{V}} p^*(u)\exp(\delta_i(u)).$$

**Step 1: Shift invariance and centring.** We perform a centring operation on $\delta_i$. Due to the shift invariance property of the softmax function, replacing $\delta_i(v)$ with $\delta_i'(v) = \delta_i(v) - C_i$ does not change $p_i(v)$. In particular, by choosing $C_i = \sum_u p^*(u)\delta_i(u)$, we have $\sum_u p^*(u)\delta_i'(u) = 0$. It remains to check that the shifted version $\delta_i'(v)$ still satisfies the assumptions (up to some constants).

- Bounded magnitude: $|\delta_i'(v)| \le |\delta_i(v)| + \sum_u p^*(u)|\delta_i(u)| \le 2\xi$;

- Zero mean: $\mathbb{E}[\delta_i'(v)] = \mathbb{E}[\delta_i(v)] - \sum_u p^*(u)\mathbb{E}[\delta_i(u)] = 0$;

- Variance over vocabulary remains unchanged:

$$\text{Var}_{u\sim p^*}(\delta_i'(u)) = \sum_u p^*(u)(\delta_i'(u))^2 = \sum_u p^*(u)\Big(\delta_i(u) - \sum_v p^*(v)\delta_i(v)\Big)^2 = \text{Var}_{u\sim p^*}(\delta_i(u)).$$

Therefore, for the remainder of the proof, we assume, without loss of generality, that $\delta_i$ satisfies the centring property:

$$\mathbb{E}_{u\sim p^*}[\delta_i(u)] = \sum_{u\in\mathcal{V}} p^*(u)\delta_i(u) = 0. \tag{7}$$

Fix a $v \in \mathcal{V}$, using triangle inequality, we can write

$$|\bar{p}_N(v) - p^*(v)| \le \left| \bar{p}_N(v) - \frac{1}{N}\sum_{i=1}^{N}\mathbb{E}[p_i(v)] \right| + \left| \frac{1}{N}\sum_{i=1}^{N}\mathbb{E}[p_i(v)] - p^*(v) \right|. \tag{8}$$

**Step 2: Concentration around the mean.** We first bound the first term in (8). Note that $p_i(v) \in [0,1]$ is bounded for $i \in \mathbb{N}$. As $\delta_1, \delta_2, \ldots$ are independent random vectors according to Assumption 2.2(b), then $p_1(v), p_2(v), \ldots$ are also independent and bounded. By Hoeffding's inequality (Hoeffding, 1963), we have

$$P\left( \left| \bar{p}_N(v) - \frac{1}{N}\sum_{i=1}^{N}\mathbb{E}[p_i(v)] \right| \ge t \right) \le 2\exp\left(-2Nt^2\right). \tag{9}$$

We study the bias term in (8) in the next steps.

**Step 3: Upper and lower bounds on $p_i(v)$.** Given Assumption 2.2(a), $|\delta_i(v)| \leq \xi \ll 1$. We use standard inequalities $1 + y \leq e^y \leq 1 + y + y^2$, valid for $|y| \leq 1$. First, we bound the normalisation quantity $Z_i$:

$$Z_i \geq \sum_u p^*(u)(1 + \delta_i(u)) = 1 + \underbrace{\sum_u p^*(u)\delta_i(u)}_{0 \text{ by } (7)} = 1, \quad \text{and}$$

$$Z_i \leq \sum_u p^*(u)(1 + \delta_i(u) + \delta_i^2(u)) = 1 + 0 + \underbrace{\sum_u p^*(u)\delta_i^2(u)}_{\mathrm{Var}_{u \sim p^*}(\delta_i(u))} = 1 + \sigma_i^2.$$

Next, we derive bounds for $p_i(v)$. First, we upper bound it by

$$p_i(v) = \frac{p^*(v)e^{\delta_i(v)}}{Z_i} \leq \frac{p^*(v)(1 + \delta_i(v) + \delta_i^2(v))}{1} = p^*(v)\left(1 + \delta_i(v) + \delta_i^2(v)\right).$$

Using the inequality $\frac{1}{1+x} \geq 1 - x$ for $x \in \mathbb{R}$, we lower bound $p_i(v)$ by

$$p_i(v) \geq \frac{p^*(v)(1 + \delta_i(v))}{1 + \mathrm{Var}_{u \sim p^*}(\delta_i(u))} \geq p^*(v)(1 + \delta_i(v))(1 - \mathrm{Var}_{u \sim p^*}(\delta_i(u)))$$
$$= p^*(v)\left(1 + \delta_i(v) - \mathrm{Var}_{u \sim p^*}(\delta_i(u)) - \delta_i(v)\mathrm{Var}_{u \sim p^*}(\delta_i(u))\right) \geq p^*(v)\left(1 + \delta_i(v) - 2\mathrm{Var}_{u \sim p^*}(\delta_i(u))\right).$$

Putting things together, we have,

$$p^*(v)\left(\delta_i(v) - 2\mathrm{Var}_{u \sim p^*}(\delta_i(u))\right) \leq p_i(v) - p^*(v) \leq p^*(v)\left(\delta_i(v) + \delta_i^2(v)\right). \tag{10}$$

**Step 4: Bounds on the bias term.** Taking the expectation on all sides of (10), we have, by Assumption 2.2(c), that

$$p^*(v)\underbrace{\mathbb{E}[\delta_i(v)]}_{0} - 2p^*(v)\mathbb{E}\left[\mathrm{Var}_{u \sim p^*}(\delta_i(u))\right] \leq \mathbb{E}[p_i(v)] - p^*(v) \leq p^*(v)\underbrace{\mathbb{E}[\delta_i(v)]}_{0} + p^*(v)\mathbb{E}[\delta_i^2(v)].$$

Using Assumption 2.2(d), we have

$$\sup_{v \in \mathcal{V}} p^*(v)\mathbb{E}\left[\mathrm{Var}_{u \sim p^*}(\delta_i(u))\right] \leq \eta^2 \quad \text{and} \quad \sup_{v \in \mathcal{V}} p^*(v)\mathbb{E}[\delta_i^2(v)] \leq \mathbb{E}\left[\sum_u p^*(u)\delta_i^2(u)\right] = \eta^2.$$

Thus

$$\sup_{v \in \mathcal{V}}\left|\mathbb{E}[p_i(v)] - p^*(v)\right| \leq 2\eta^2. \tag{11}$$

**Step 5: Final derivation.** Combining (9) and (11), we conclude that for any $\delta > 0$, with probability at least $1 - \delta$,

$$\left|\bar{p}_N(v) - p^*(v)\right| \leq \sqrt{\frac{2\log(2/\delta)}{N}} + 2\eta^2.$$

Note that this holds for any fixed $v \in \mathcal{V}$. For the $\ell_\infty$ distance, by a standard union bound argument, with probability at least $1 - \delta$,

$$\left\|\bar{p}_N(\cdot|x) - p^*(\cdot|x)\right\|_\infty = \sup_{v \in \mathcal{V}}\left|\bar{p}_N(v) - p^*(v)\right|$$
$$\leq \sup_{v \in \mathcal{V}}\left|\bar{p}_N(v) - \frac{1}{N}\sum_{i=1}^N \mathbb{E}[p_i(v)]\right| + \sup_{v \in \mathcal{V}}\left|\frac{1}{N}\sum_{i=1}^N \mathbb{E}[p_i(v)] - p^*(v)\right|$$
$$\leq \sqrt{\frac{2\log(2|\mathcal{V}|/\delta)}{N}} + 2\eta^2.$$

$\square$

# B. Extension of the Theoretical Result to Grouped Watermarking Settings

The theoretical analysis in Section 2 assumes full independence across all $N$ providers. In practice, however, certain providers may share common watermarking toolkits, licensing agreements, or underlying model families, inducing statistical dependence among their perturbation vectors.

We show here that the convergence guarantee extends naturally to a *grouped* setting in which providers are partitioned into independent clusters, with only conditional independence required within each cluster. We also relax the unbiasedness assumption by allowing models within a group to share a common perturbation component.

**Assumption B.1** (Grouped Perturbations)**.** Consider a set of $N$ providers. Suppose they can be partitioned into $M$ groups:

$$G_1, \ldots, G_M, \quad \bigcup_{g=1}^{M} G_g = \{1, \ldots, N\}, \quad G_g \cap G_{g'} = \emptyset \text{ for } g \neq g',$$

with $\sum_{g=1}^{M} |G_g| = N$. For each group $g$, let $W_g$ be a group-level latent variable. Each provider is associated with a random perturbation vector $\delta_i(\cdot, x)$ that modulates the output distribution. For every context $x \in \mathcal{X}$, we assume the following properties hold for $\{\delta_i(\cdot, x)\}_{i=1}^{N}$:

**(a) Bounded Magnitude:** The perturbation magnitude is uniformly bounded by a constant $\xi \leq 1$. Specifically, $\|\delta_i(\cdot, x)\|_\infty \leq \xi$ for all providers $i$.

**(b) Group independence structure:** *Across groups*, the collections $\left(\{\delta_i(\cdot, x)\}_{i \in G_g}, W_g\right)_{g=1,\ldots,M}$ are mutually independent; *within each group*, conditional on $W_g$, the perturbations $\{\delta_i(\cdot, x)\}_{i \in G_g}$ are mutually independent. Specifically, these two assumptions imply

$$\mathbb{P}\left(\bigcap_{i=1}^{N} \{\delta_i(\cdot, x) \in A_i\} \,\Big|\, W_1, \ldots, W_M\right) = \prod_{g=1}^{M} \prod_{i \in G_g} \mathbb{P}\left(\delta_i(\cdot, x) \in A_i \mid W_g\right)$$

for any measurable sets $A_1, \ldots, A_N$.

**(c) Group-specific Bias:** For each $g \in \{1, \ldots, M\}$ and $i \in G_g$,

$$\mathbb{E}[\delta_i(\cdot, x) \mid W_g] = b_g(\cdot, x),$$

where $b_g(\cdot, x)$ is a group-specific bias function.

**(d) Conditionally Bounded Expected Variance of the Idiosyncratic Perturbation:** For each $g \in \{1, \ldots, M\}$ and $i \in G_g$, we decompose the perturbation into a group-specific component and an idiosyncratic component as

$$\delta_i(\cdot, x) = b_g(\cdot, x) + \varepsilon_i(\cdot, x),$$

where $b_g(\cdot, x)$ is defined above and shared by all providers within group $G_g$, and $\varepsilon_i(\cdot, x)$ denotes the idiosyncratic perturbation. The conditional expectation of the weighted variation of the idiosyncratic perturbations across the vocabulary (weighted by the consensus probability) is bounded by a constant $\eta^2$. That is,

$$\mathbb{E}\left[\mathrm{Var}_{u \sim p^*}\left(\varepsilon_i(u, x)\right) \mid W_g\right] \leq \eta^2$$

where

$$\mathrm{Var}_{u \sim p^*}(\varepsilon_i(u, x)) := \sum_u p^*(u|x)\left(\varepsilon_i(u, x) - \sum_v p^*(v|x)\varepsilon_i(v, x)\right)^2.$$

Define the group consensus distribution for group $g$ as

$$p_g^\dagger(v|x) := \frac{p^*(v|x) \exp(b_g(v, x))}{\sum_{u \in \mathcal{V}} p^*(u|x) \exp(b_g(u, x))}, \tag{12}$$

and the group-size-weighted average $\bar{p}^\dagger(\cdot|x) := \frac{1}{N} \sum_{g=1}^{M} n_g p_g^\dagger(\cdot|x)$. The irreducible group bias is defined as

$$B(x) := \left\|\bar{p}^\dagger(\cdot|x) - p^*(\cdot|x)\right\|_\infty = \sup_{v \in \mathcal{V}} \left|\bar{p}^\dagger(v|x) - p^*(v|x)\right|. \tag{13}$$

**Theorem B.2.** *Under Assumption B.1, for any fixed context $x$, let $\bar{p}_N(\cdot|x) = \frac{1}{N}\sum_{i=1}^{N} p_i(\cdot|x)$ be the aggregated distribution. Then, for any $\delta > 0$, with probability at least $1 - \delta$,*

$$\left\|\bar{p}_N(\cdot \mid x) - p^*(\cdot \mid x)\right\|_\infty \lesssim \sqrt{\frac{\log(|\mathcal{V}|/\delta)}{N}} + \eta^2 + B(x).$$

*Proof.* For brevity, we again omit the dependency on $x$ in the notation throughout this proof. Recall that the perturbed distribution for model $i$ is given by

$$p_i(v) = \frac{p^*(v)\exp(\delta_i(v))}{\sum_{u\in\mathcal{V}} p^*(u)\exp(\delta_i(u))} = \frac{p^*(v)\exp(b_g(v) + \varepsilon_i(v))}{\sum_{u\in\mathcal{V}} p^*(u)\exp(b_g(u) + \varepsilon_i(u))} = \frac{p_g^\dagger(v)\exp(\varepsilon_i(v))}{\sum_u p_g^\dagger(u)\exp(\varepsilon_i(u))}, \tag{14}$$

where $p_g^\dagger(\cdot)$ is defined in (12). Denote

$$\mu_W(v) := \frac{1}{N}\sum_{g=1}^{M}\sum_{i\in G_g} \mathbb{E}[p_i(v) \mid W_g].$$

By the triangle inequality,

$$\|\bar{p}_N - p^*\|_\infty \leq \underbrace{\|\bar{p}_N - \mu_W\|_\infty}_{\text{Concentration}} + \underbrace{\|\mu_W - \bar{p}^\dagger\|_\infty}_{\text{Second-order idiosyncratic effect}} + \underbrace{\|\bar{p}^\dagger - p^*\|_\infty}_{\text{group bias } B(x)}. \tag{15}$$

The third term equals $B(x)$ by definition (13).

**Step 1: Concentration around the mean.** We now bound the first term. By Assumption B.1(b), for $i \in G_g$, the conditional law of $\delta_i$ given $W$ depends only on $W_g$. Thus, we have

$$\mathbb{E}[p_i(v) \mid W] = \mathbb{E}[p_i(v) \mid W_g] \qquad \text{and} \qquad \mu_W(v) = \frac{1}{N}\sum_{i=1}^{N}\mathbb{E}[p_i(v) \mid W].$$

In addition, conditional on $W = (W_1, \ldots, W_M)$, the perturbations $\delta_1, \ldots, \delta_N$ are mutually independent. Since $p_i(v)$ is a measurable function of $\delta_i$, the random variables $p_1(v), \ldots, p_N(v)$ are also mutually independent conditional on $W$. As $p_i(v) \in [0, 1]$ is bounded for $i \in \mathbb{N}$. By Hoeffding's inequality (Hoeffding, 1963), we have

$$\mathbb{P}\big(|\bar{p}_N(v) - \mu_W(v)| \geq t \mid W\big) \leq 2\exp(-2Nt^2).$$

Taking expectation over $W$, the same inequality holds unconditionally. A union bound over $v \in \mathcal{V}$ then yields: with probability at least $1 - \delta$,

$$\|\bar{p}_N - \mu_W\|_\infty \leq \sqrt{\frac{\log(|\mathcal{V}|/\delta)}{2N}}. \tag{16}$$

**Step 2: Shift invariance and centering.** Denote $\widetilde{\varepsilon}_i(v) := \varepsilon_i(v) - \sum_u p_g^\dagger(u)\varepsilon_i(u)$, which ensures

$$\sum_u p_g^\dagger(u)\widetilde{\varepsilon}_i(u) = 0. \tag{17}$$

By shift invariance of the softmax function, we have

$$p_i(v) = \frac{p_g^\dagger(v)\exp(\widetilde{\varepsilon}_i(v))}{\sum_u p_g^\dagger(u)\exp(\widetilde{\varepsilon}_i(u))}.$$

By Assumption B.1(c) and the definition of the group consensus (12), we have that $p_g^\dagger$ is $W_g$-measurable and thus, for every $v \in \mathcal{V}$

$$\mathbb{E}[\widetilde{\varepsilon}_i(v) \mid W_g] = \mathbb{E}[\varepsilon_i(v) \mid W_g] - \sum_{u\in\mathcal{V}} \mathbb{E}[p_g^\dagger(u)\varepsilon_i(u) \mid W_g]$$

$$= \underbrace{\mathbb{E}[\varepsilon_i(v) \mid W_g]}_{0} - \sum_{u\in\mathcal{V}} p_g^\dagger(u)\underbrace{\mathbb{E}[\varepsilon_i(u) \mid W_g]}_{0} = 0.$$

**Step 3: Bounding the second-order idiosyncratic effect term.** We write

$$\|\mu_W - \bar{p}^\dagger\|_\infty = \sup_{v \in \mathcal{V}} \left| \frac{1}{N} \sum_{g=1}^{M} \sum_{i \in G_g} \left( \mathbb{E}[p_i(v) \mid W_g] - p_g^\dagger(v) \right) \right| \leq \frac{1}{N} \sum_{g=1}^{M} \sum_{i \in G_g} \sup_{v \in \mathcal{V}} \left| \mathbb{E}[p_i(v) \mid W_g] - p_g^\dagger(v) \right|$$

Let $R_i(v) := e^{\widetilde{\varepsilon}_i(v)} - 1 - \widetilde{\varepsilon}_i(v)$ and $A_i := \sum_u p_g^\dagger(u) R_i(u)$. Then, using (17), we have

$$p_i(v) - p_g^\dagger(v) = \frac{p_g^\dagger(v)(1 + \widetilde{\varepsilon}_i(v) + R_i(v))}{1 + A_i} - p_g^\dagger(v) = p_g^\dagger(v)\widetilde{\varepsilon}_i(v) + p_g^\dagger(v)\frac{R_i(v) - A_i - A_i\widetilde{\varepsilon}_i(v)}{1 + A_i}.$$

By the result of Step 2,

$$\mathbb{E}[p_i(v) \mid W_g] - p_g^\dagger(v) = \underbrace{p_g^\dagger(v)\mathbb{E}[\widetilde{\varepsilon}_i(v) \mid W_g]}_{=0} + \mathbb{E}\left[ p_g^\dagger(v)\frac{R_i(v) - A_i - A_i\widetilde{\varepsilon}_i(v)}{1 + A_i} \ \middle| \ W_g \right].$$

Since $R_i(v) = e^{\widetilde{\varepsilon}_i(v)} - 1 - \widetilde{\varepsilon}_i(v) \geq 0$ for all $v$, we have $A_i = \sum_u p_g^\dagger(u) R_i(u) \geq 0$ and thus

$$p_g^\dagger(v) \left| \frac{R_i(v) - A_i - A_i\widetilde{\varepsilon}_i(v)}{1 + A_i} \right| \leq p_g^\dagger(v) \left( |R_i(v)| + |A_i| + |A_i\widetilde{\varepsilon}_i(v)| \right).$$

By Assumption B.1(a), $\|\delta_i\|_\infty \leq \xi$. As $\varepsilon_i(v) = \delta_i(v) - \mathbb{E}[\delta_i(v)|W_g]$, we have $\|\varepsilon_i\|_\infty \leq 2\xi$. It follows that $\|\widetilde{\varepsilon}_i\|_\infty \leq 4\xi$. Then we have $|R_i(v)| = |e^{\widetilde{\varepsilon}_i(v)} - 1 - \widetilde{\varepsilon}_i(v)| \leq C_\xi \widetilde{\varepsilon}_i(v)^2$ for all $v \in \mathcal{V}$, where $C_\xi$ is a constant depending only on $\xi$, and hence $|A_i| = \left| \sum_u p_g^\dagger(u) R_i(u) \right| \leq \sum_u p_g^\dagger(u) |R_i(u)| \leq C_\xi \sum_u p_g^\dagger(u) \widetilde{\varepsilon}_i(u)^2 = C_\xi \mathrm{Var}_{u \sim p_g^\dagger}(\varepsilon_i(u))$. Using these bounds, we obtain

$$p_g^\dagger(v) \left| \frac{R_i(v) - A_i - A_i\widetilde{\varepsilon}_i(v)}{1 + A_i} \right| \leq C_\xi' \mathrm{Var}_{u \sim p_g^\dagger}(\varepsilon_i(u)).$$

and hence

$$\sup_{v \in \mathcal{V}} \left| \mathbb{E}[p_i(v) \mid W_g] - p_g^\dagger(v) \right| \leq C_\xi' \mathbb{E}\left[ \mathrm{Var}_{u \sim p_g^\dagger}(\varepsilon_i(u)) \mid W_g \right]. \tag{18}$$

We next relate $\mathrm{Var}_{u \sim p_g^\dagger}(\varepsilon_i(u))$ to $\mathrm{Var}_{u \sim p^*}(\varepsilon_i(u))$. Since $|b_g(u)| \leq \xi$, on the support of $p^*$ we have

$$\frac{p_g^\dagger(u)}{p^*(u)} = \frac{\exp(b_g(u))}{\sum_{z \in \mathcal{V}} p^*(z)\exp(b_g(z))} \leq \frac{e^\xi}{e^{-\xi}} = e^{2\xi}.$$

Therefore,

$$\mathrm{Var}_{u \sim p_g^\dagger}(\varepsilon_i(u)) = \inf_{a \in \mathbb{R}} \left\{ \sum_{u \in \mathcal{V}} p_g^\dagger(u)(\varepsilon_i(u) - a)^2 \right\} \leq \sum_{u \in \mathcal{V}} p_g^\dagger(u) \left( \varepsilon_i(u) - \sum_{z \in \mathcal{V}} p^*(z)\varepsilon_i(z) \right)^2$$

$$\leq e^{2\xi} \sum_{u \in \mathcal{V}} p^*(u) \left( \varepsilon_i(u) - \sum_{z \in \mathcal{V}} p^*(z)\varepsilon_i(z) \right)^2 = e^{2\xi} \mathrm{Var}_{u \sim p^*}(\varepsilon_i(u)).$$

Taking conditional expectations and using Assumption B.1(d), we obtain

$$\mathbb{E}\left[ \mathrm{Var}_{u \sim p_g^\dagger}(\varepsilon_i(u)) \middle| W_g \right] \leq e^{2\xi} \mathbb{E}\left[ \mathrm{Var}_{u \sim p^*}(\varepsilon_i(u)) | W_g \right] \leq e^{2\xi} \eta^2.$$

Substituting this into (18), we then have $\sup_{v \in \mathcal{V}} \left| \mathbb{E}[p_i(v) \mid W_g] - p_g^\dagger(v) \right| \leq C_\xi' e^{2\xi} \eta^2$. Averaging over all providers gives

$$\|\mu_W - \bar{p}^\dagger\|_\infty \leq C_\xi' e^{2\xi} \eta^2. \tag{19}$$

**Step 4: Final derivation.** Substituting (16), (19), and the definition of $B(x)$ into (15), we conclude that with probability at least $1 - \delta$,

$$\|\bar{p}_N - p^*\|_\infty \leq \sqrt{\frac{\log(|\mathcal{V}|/\delta)}{2N}} + C'_\xi e^{2\xi}\eta^2 + B(x) \lesssim \sqrt{\frac{\log(|\mathcal{V}|/\delta)}{N}} + \eta^2 + B(x).$$

□

## C. Robustness to Biased Watermark Perturbations

Assumption 2.2(c) assumes watermark perturbations as zero-mean around the consensus distribution. We stress-test this condition by deploying a biased green-red-list watermark that consistently promotes a fixed set of tokens with a bias. We evaluate the watermarked baseline and WASH with both the accuracy on the GSM8K task, and the watermark detection z-score.

Table 8 shows a utility-detection trade-off: stronger bias makes the baseline easier to detect, but rapidly hurts task performance. This further validates the practical plausibility of Assumption 2.2(c): rational providers have no incentive to deploy systematically biased perturbations, as the severe accuracy degradation (from 0.443 to 0.023) demonstrates that deviating from the zero-mean condition directly harms generation quality, undermining their own service. WASH remains effective across all bias values, keeping detector z-scores below the no-detection threshold ($z \leq 4$) and improving accuracy over the corresponding biased baseline.

This is consistent with the theoretical prediction in Appendix B when perturbations carry a shared bias, an irreducible term $B(x)$ persists in the convergence bound, explaining why the detection z-scores under WASH do not vanish completely but remain below the detection threshold.

*Table 8.* Robustness to biased watermark perturbations.

| Bias | Watermarked Accuracy ↑ | Watermarked Detection $z$ ↓ | WASH Accuracy ↑ | WASH Detection $z$ ↓ |
|------|------------------------|------------------------------|-----------------|----------------------|
| 2.0  | 0.443                  | 3.71                         | 0.563           | 1.35                 |
| 4.0  | 0.137                  | 11.49                        | 0.507           | 1.92                 |
| 6.0  | 0.017                  | 15.55                        | 0.403           | 2.65                 |
| 8.0  | 0.010                  | 15.67                        | 0.253           | 2.93                 |
| 10.0 | 0.023                  | 15.76                        | 0.220           | 2.95                 |

## D. Experiment Details

### D.1. Implementation Details

**Detection thresholds.** For *generation-time attacks*, we follow the z-score protocol of Liu et al. (2025) to quantify watermark signal strength. Following the original definition, Detection confidence is categorised as high-confidence identification ($z > 10$), low-confidence identification ($4 < z \leq 10$), and no detection ($z \leq 4$).

For *final-text rewrite attacks*, perturbation detection on the small set of generated tokens is no longer compatible, so we additionally use native sequence detectors (Pan et al., 2024). Following Liu et al. (2025), we generate the sequence to be detected on the C4 dataset (Raffel et al., 2020): we truncate each sample to 30 tokens as the prompt, generate 200 additional tokens with the watermarked model, and then perform detection on the generated sequence. Since z-score magnitudes vary substantially across watermarking schemes, directly comparing z-scores across schemes is not meaningful. We therefore report the removal effect using TPR@5% FPR, the true-positive rate of identifying a watermarked sequence at a threshold calibrated to falsely flag 5% of unwatermarked sequences, which is widely used as a watermark robustness measurement due to its consistency across watermark types, text types, and lengths (Kirchenbauer et al., 2023b). Lower TPR@5% FPR indicates stronger removal. We categorise a strong watermark signal for TPR@5% FPR $\geq 75\%$, while a low-confidence detection with $50 \leq$ TPR@5% FPR $\leq 75\%$.

### D.2. Detailed Experiment Results

Table 9 shows the detailed experiment results for Figure 3(a) and (b), the watermark detection evaluation with mixing diverse watermark schemes and base models. For the fixed-base model setting, we randomly sampled 15 ensemble combinations for

each mixture amount $N$ and ran the detection task 5 times each to obtain stable z-scores. For the mixed-base model setting, we experimented with a larger range of $N$ from 1 to 8, and sampled 50 mixture combinations for each $N$ due to the large sampling space. The results follow the scaling law: with a larger ensemble size, the watermark signal decays more.

For the fixed-base model setting (Figure 3(a)), the detection signal diminishes rapidly as $N$ increases, with the Llama3.1-8B model dropping from an extremely high z-score ($\approx 150$) to $\approx 10$ at $N = 5$. This empirical decay mirrors our theoretical prediction of $O(1/\sqrt{N})$. However, we observe that the signal does not vanish as completely as in the heterogeneous setting. This residual signal likely arises because the shared systematic bias ($\delta_{sys}$) of the identical base model persists across the ensemble, hindering the complete cancellation of artefacts. More importantly, this single-model scenario is rarely available in practice, as providers seldom expose multiple watermarked versions of the same model to end users.

*Table 9.* Experiment results with a mixture of watermarks and base models for Figure 3(a) and (b). ▇ indicates high-confidence watermark identification, and ▇ indicates low-confidence watermark identification, while no colour indicates no watermark identified.

| Base Model | Mixture Amount | Mean Z-score | Std Z-score | Lower Bound | Upper Bound |
|---|---|---|---|---|---|
| | 1 | 147.352 | 107.821 | 92.787 | 201.917 |
| | 2 | 27.417 | 24.890 | 14.816 | 40.018 |
| **Llama3.1-8B** | 3 | 20.511 | 18.855 | 10.969 | 30.053 |
| | 4 | 15.890 | 15.370 | 8.111 | 23.668 |
| | 5 | 9.344 | 8.935 | 4.822 | 13.865 |
| | 1 | 24.304 | 27.525 | 10.374 | 38.233 |
| | 2 | 6.934 | 6.305 | 3.744 | 10.125 |
| **Qwen3-8B** | 3 | 3.308 | 4.266 | 1.150 | 5.467 |
| | 4 | 1.650 | 2.862 | 0.201 | 3.098 |
| | 5 | 1.716 | 1.284 | 1.066 | 2.366 |
| | 1 | 35.724 | 44.901 | 13.001 | 58.447 |
| | 2 | 12.491 | 14.270 | 5.269 | 19.713 |
| **Ministral3-8B** | 3 | 3.241 | 3.758 | 1.339 | 5.143 |
| | 4 | 1.201 | 2.143 | 0.117 | 2.286 |
| | 5 | 0.463 | 0.965 | -0.024 | 0.951 |
| | 1 | 59.909 | 76.434 | 38.722 | 81.095 |
| | 2 | 14.956 | 15.020 | 10.793 | 19.120 |
| | 3 | 3.866 | 7.403 | 1.815 | 5.918 |
| | 4 | 1.878 | 5.510 | 0.350 | 3.405 |
| **Mixed Model** | 5 | 0.087 | 3.666 | -0.929 | 1.103 |
| | 6 | -0.066 | 2.427 | -0.739 | 0.606 |
| | 7 | -0.344 | 2.174 | -0.947 | 0.259 |
| | 8 | -0.342 | 2.669 | -1.082 | 0.398 |

# E. Fluency-Aware Routing Example

Figure 4 shows two complete generation flows for a text completion task in the C4 dataset and a reasoning task on the MMLU Law subset, including the Fluency-Aware Routing triggered in between. These generations are conducted by a mixture of three watermarked models: Llama-Aar, Ministral-DIPMark, and Qwen-KGW.

The routing mechanism is triggered mostly on words that are tokenised differently across models. For example, in the MMLU Law task, the word "negligent" is tokenised to ['negl', 'igent'] by Ministral, but is maintained as a complete word token by Llama and Qwen. During Fluency-Aware Routing, the span is completed using only the specialist models that share the same tokenisation scheme for the target span. After routing, the span is re-synchronised by each model's own tokeniser, ensuring that the span's understanding won't be damaged by misaligned tokenisations.

---

**Case study: Fluency-Aware Routing**

**Task.** C4 Completion

**Prefix.** *To Be Built By Professional Local Builders! Post Modern Located On The Vanderbilt Little Neck Peninsula In Centerport. SD#6, Open Floor Plan Perfect For Entertaining With*

**Completion.** Open Kitchen, Family Room, **Dining** ← [' Dining'] And Bar Area. Private Master **Bedroom** ← [' Bedroom'], Two Additional En-Suite **Bedrooms** ← [' Bedrooms'] And One Additional Bed Or Office. All Bedrooms With Walk-In Closets And Bathrooms With Dual Sinks warmer. 1st Likewise With Powder Room, **2nd** ← ['2', 'nd'] With **Laundry.** ← [' Laundry', '.'] **2.5** ← ['2', '.', '5'] car garage. **15** ← ['1', '5'] min to Commuter **Parking** ← [' Parking'] and **LIRR** ← ['IRR']

---

**Task.** MMLU Law Reasoning

**Reasoning.** The fan was **55** ← ['5', '5']% **negligent** ← [' negligent'] (speeding) and the driver was **45** ← ['4', '5']% **negligent** ← ' negligent' (red light violation). The friend was not found **negligent.** ← [' negl', 'igent', '.'] Under modified comparative **negligence** ← [' negl', 'igence'], if the plaintiff is not more than **50** ← ['5', '0']% **negligent** ← [' negl', 'igent'], they can recover damages, but they are entitled to a percentage of the damages according to their relative **negligence.** ← [' negligence', '.'] However, the question states that the jury found $**100,000** ← ['100', ',', '000'] in damages. If the fan was blooded, and the friend suffered $**100** ← ['1', '0', '0'],**000** ← ['000'] in damages, but the friend's **negligence** ← [' negl', 'igence'] was only 0% (not found **negligent** ← [' negligent']), then the friend can recover the full $**100** ← ['1', '0', '0'],**000.** ← ['000', '.'] The answer is A.

---

*Figure 4.* Examples of Fluency-Aware Routing on C4 completion and MMLU Law reasoning tasks. The routed spans are shown in bold blue, with the list of routed tokens appended.

