# OpenReview forum: "Linear Ensembles Wash Away Watermarks: On the Fragility of Distributional Perturbations in LLMs"
_ICML.cc/2026/Conference — ICML 2026 regular_

### Official Review · Reviewer_Kbpp · 2026-03-07

**Soundness:** 3
**Presentation:** 3
**Significance:** 2
**Originality:** 3
**Overall Recommendation:** 4
**Confidence:** 2

**Summary:**

This paper demonstrates that watermarks embedded in AI-generated text can be neutralized through an ensemble technique that utilizes multiple AI models simultaneously.
The paper focused on the fact that models different providers use independent statistical modulations. The paper demonstrated, both mathematically and experimentally, that averaging the output probability distributions causes the watermark signals to cancel out.
To implement this, they developed the WASH (Watermark Attenuation via Statistical Hybridisation) framework, which resolves discrepancies between models with different tokenization methods.

**Compliance With Llm Reviewing Policy:**

Affirmed.

**Final Justification:**

After carefully consideration of the rebuttal, I have updated my score.
The authors have adequately addressed all three of my concerns with additional empirical evidence.
- The VRAM overhead
- Domain-specific vulnerability
- Creative generation quality

**Key Questions For Authors:**

- Given the extreme memory bottlenecks of loading multiple large language models simultaneously, can the WASH framework genuinely be considered practical and deployable for average users or small enterprises without high-end GPU clusters?

- Have you evaluated the WASH framework's vulnerability on highly domain-specific texts (e.g., medical or legal documents)? How can you guarantee that the watermark remains undetectable when frequent vocabulary mismatches force prolonged reliance on a single specialist model, thereby risking significant watermark accumulation?

- Your quality evaluation relies entirely on deterministic, fact-based benchmarks. Given that continuous probability blending across multiple models risks diluting stylistic diversity and producing "bland" text, how can you guarantee that the WASH framework preserves the stylistic nuances and natural fluency required for open-ended, creative generation tasks?

**Limitations:**

yes

**Strengths And Weaknesses:**

### Strength
- The paper introduces a novel method to neutralize watermarks across six different SOTA watermarking schemes (e.g., KGW, DIPMark, Aar..)
- By employing "flency-aware routing" and "context re-synchronisation", the paper maintains strong semantic integrity compared to previous methods.
- The paper goes beyond empirical observations by providing a rigorous mathematical proof
### Weakness
- The paper claims computational efficiency based solely on inference time, but severely overlooks the massive GPU memory (VRAM) overhead required to run 3 to 5 distinct 8B models in parallel with synchronized KV caching.
- The paper claims that the temporary reappearance of watermark signals during Fluency-Aware Routing is too fragmented to be detected. However, in highly technical or specialized domains with frequent out-of-vocabulary terms, the system may be forced to rely on a single specialist model for extended periods, potentially allowing the watermark signal to accumulate to strongly detectable levels.

---

> ### Author Rebuttal · Authors · 2026-03-31
>
> > **Q1**. The paper claims computational efficiency based solely on inference time, but severely overlooks the massive GPU memory (VRAM) overhead required to run 3 to 5 distinct 8B models in parallel with synchronized KV caching. Given the extreme memory bottlenecks of loading multiple large language models simultaneously, can the WASH framework genuinely be considered practical and deployable for average users or small enterprises without high-end GPU clusters?
>
> We thank the reviewer for this important point. We agree that our original efficiency discussion emphasises latency but does not separately report VRAM cost. Since WASH uses multi-model ensembling, there is a clear **memory-latency trade-off**. To support different resource budgets, we provide two deployment variants:
>
> - **WASH-Parallel**: parallel multi-model decoding with synchronised KV caches, with $O(N)$ GPU memory cost and $O(1)$ per-token latency;
>
> - **WASH-Sequential**: sequential multi-model decoding on a single GPU with memory reuse, with $O(1)$ GPU memory cost and $O(N)$ per-token latency.
>
> We report the empirical results below with a blending size of 3 for long-text generation (512 tokens) using bf16 models:
>
> | Method Variant | Peak VRAM | Per-token Latency |
> |-|-:|-:|
> | Unwatermarked | 15.09 GB (1.00x) | 29.5 ms (0.98x) |
> | Watermarked     | 15.10 GB (1.00x) | 30.0 ms (1.00x) |
> | De-mark             | 16.58 GB (1.10×) | 190.6 ms (6.35×) |
> | ToBlend              | 71.46 GB (4.73×) | 172.3 ms (5.74×) |
> | WASH-Parallel   | 40.87 GB (2.71×) | 56.9 ms (1.90×) |
> | WASH-Sequential | 15.76 GB (1.04×) | 165.5 ms (5.52×) |
>
> These results confirm the trade-off discussed above. Also, under both deployment regimes, WASH remains more efficient than the prior removal baselines.
>
> > **Q2**. Have you evaluated the WASH framework's vulnerability on highly domain-specific texts (e.g., medical or legal documents)? How can you guarantee that the watermark remains undetectable when frequent vocabulary mismatches force prolonged reliance on a single specialist model, thereby risking significant watermark accumulation?
>
> We thank you for this suggestion. We agree that highly domain-specific text is a meaningful stress test for WASH.  In WASH, routing happens only when the next sampled token is not in the shared vocabulary. The prolonged reliance on a single model is unlikely to happen, since
>
> - During routing, WASH does not deterministically commit to a single model; instead, it samples from the set of compatible specialist models that are able to complete the current word.
>
> - Once the current word is completed (which won’t typically last long), all models are re-synchronised and the usual ensemble decoding resumes.
>
> To further address the reviewer’s concern, we add experiments on the medical and legal subsets of MMLU and report average detection scores. In addition, we report routing diagnostics to quantify whether domain-specific vocabulary induces sustained single-model dependence.
>
> | Domain | Routed-token fraction | Avg. routed span length| Avg. detection scores|
> |-|-:|-:|-:|
> | Medical | 3.2% | 3.7 | 0.84 |
> | Legal     | 2.7% | 3.6 | 1.08 |
>
> Across both domains, WASH maintains low detection scores, while routed spans remain short and fragmented, with singleton routing occurring only rarely.
>
> In fact, we observe that routing is more likely to be triggered by standard terminology composed of simple subword units, such as “dissection”, rather than more complex terms, as the latter are typically split into widely shared subwords (e.g., “retroperitoneal” into “retro,” “per,” “itone,” “al”), reducing the likelihood of distinct routing signals.
>
> > **Q3**. Your quality evaluation relies entirely on deterministic, fact-based benchmarks. Given that continuous probability blending across multiple models risks diluting stylistic diversity and producing "bland" text, how can you guarantee that the WASH framework preserves the stylistic nuances and natural fluency required for open-ended, creative generation tasks?
>
> We thank the reviewer for this important concern. To directly evaluate open-ended creative generation, we add experiments on **WritingBench**, a benchmark designed to evaluate LLMs across 6 core writing domains.
>
> | Method               | Style | Format | Per-token Latency |
> | - | -: | -: | -: |
> | Unwatermarked  | 4.02 |    4.13   | 0.92×   |
> | Watermarked      | 4.03 |    4.17   | 1.00×   |
> | De-mark              | 3.88 |    4.20   | 43.98× |
> | ToBlend               | 2.18 |    2.45   | 9.08×   |
> | WASH                | 4.38 |    4.13   | 1.69×   |
>
> Our results show that WASH remains competitive with the unwatermarked model and consistently outperforms prior removal baselines, suggesting that multi-model averaging does not materially wash out stylistic nuance or natural fluency on the evaluated creative-writing setting.

---

> > ### Author Rebuttal · Reviewer_Kbpp · 2026-04-04
> >
> > Thank you for the detailed rebuttal and additional experiments The authors have satisfactorily addressed all three of my concerns.
> > I would encourage the authors to incorporate the rebuttal experiments into the final version of the paper.
> > I have updated my score.

---

> > > ### Author Response · Authors · 2026-04-05
> > >
> > > Thank you very much for your constructive feedback throughout the review process, which has genuinely helped us strengthen the paper. We are glad that our responses have helped address your concerns and we will incorporate the additional experiments on memory, domain-specific texts, and creative writing into the updated version of the paper.

---

### Official Review · Reviewer_SfU4 · 2026-03-10

**Soundness:** 3
**Presentation:** 2
**Significance:** 2
**Originality:** 3
**Overall Recommendation:** 3
**Confidence:** 3

**Summary:**

This paper reveals a fundamental vulnerability in watermarking mechanisms for large language models (LLMs) within a multi-model integration environment. The authors point out that when users have access to multiple independent watermarked models, a simple averaging of the output probability distributions can counteract the statistically imperceptible perturbations independently added by each provider, thereby recovering a distribution close to the original, watermark-free distribution.

**Compliance With Llm Reviewing Policy:**

Affirmed.

**Final Justification:**

Although the rebuttal has partially resolved my concerns, I still believe that the novelty and overall contribution of the paper are relatively limited and I maintain my score.

**Key Questions For Authors:**

1. How robust is the framework if the supplier adopts a defensive strategy (e.g., intentionally introducing asymmetric perturbations or nonlinear watermarks)?
2. add a quantitative analysis of the attack's costs and benefits in resource-constrained scenarios.

**Limitations:**

yes

**Strengths And Weaknesses:**

1. Formally proved for independent unbiased watermarking schemes.
2. The WASH framework is proposed, which includes "Fluency-Aware Routing" and a "Context Resynchronization" mechanism to overcome the problems of small vocabulary intersection and inconsistent granularity between different models.

Weaknesses:

1. While experiments (Figure 3c) demonstrate that collaborative watermarking can resist such attacks, the paper primarily focuses on the attack surface.
2. While WASH optimizes speed through parallel inference and caching, integrating N models inevitably consumes N times the memory and computational resources.

3. The theoretical proof relies on Assumption 2.2 (unbiasedness, independence, etc.). To what extent can the framework withstand a scenario where the supplier implements a defensive strategy.

---

> ### Author Rebuttal · Authors · 2026-03-31
>
> > **Q1**. The theoretical proof relies on Assumption 2.2 (unbiasedness, independence, etc.).
>
> The i.i.d. and zero-mean assumptions are mainly for simplicity of analysis. In fact, the identical distribution part of the assumption is not needed: the proof only uses it to write the mean as $E[p_1(v)]$, and the argument extends directly by replacing this with the average expectation across models.
>
> For the non-independence case, the result can be extended to a grouped setting, which is more realistic since models may share common watermarking mechanisms or latent factors. In this grouped setting, under conditional independence and unbiasedness within each group, the exact same convergence result holds without modification, with $N^{-1/2}$ convergence to the consensus distribution up to a second-order $η^2$ term.
>
> If models within a group share a common perturbation component with non-zero mean, an additional bias term appears; this is somewhat unavoidable since averaging cannot remove shared structure. However, when perturbations are weak, this term is typically small and aggregation still yields distributions close to p*.
>
> > **Q2**. How robust is the framework if the supplier adopts a defensive strategy (e.g., intentionally introducing asymmetric perturbations or nonlinear watermarks)?
>
> We agree that a supplier could defend against WASH by intentionally violating the assumptions that make averaging effective.
>
> **Breaking independence.** Figure 3(c) studies a coordinated-watermark control, where watermark signals are synchronised across models. In this setting, the washing effect is weakened, showing that collaboratively designed perturbations can make ensembling-based removal more difficult. This is consistent with the theory: averaging is effective against provider-specific perturbations, but cannot fully remove shared structure.
>
> **Breaking unbiasedness.** We additionally introduce a simple **biased watermark** as a stress test. Concretely, for every context, we designate a fixed greenlist and increase the probabilities of greenlist tokens by $\delta$, while leaving redlist tokens unchanged and then renormalising. This induces a persistent positive bias of strength $\delta$. As expected, stronger bias makes watermark removal more difficult. However, it also degrades generation quality of the watermarked model, since the watermark persistently pushes probability mass toward pre-selected greenlist tokens even when those tokens are semantically inappropriate. This will force providers to keep such bias small.
>
>
> | Bias strength $\delta$ | Watermarked Accuracy ↑ | Watermarked Detect score ↑ | WASH Accuracy ↑ | WASH Detect score ↓ |
> |-|-:|-:|-:|-:|
> | 2.0    | 44.33 |   3.71 | 56.33 | 1.35 |
> | 4.0    | 13.67 | 11.49 | 50.67 | 1.92 |
> | 6.0    |   1.67 | 15.55 | 40.33 | 2.65 |
> | 8.0    |   1.00 | 15.67 | 25.33 | 2.93 |
> | 10.0  |   2.33 | 15.76 | 22.00 | 2.95 |
>
> As discussed in the Impact Statement, our results suggest that **provider collaboration** can itself serve as a defense, improving robustness without requiring overly strong bias that would substantially degrade generation quality.
>
> > **Q3**. While WASH optimizes speed through parallel inference and caching, integrating N models inevitably consumes N times the memory and computational resources. Add a quantitative analysis of the attack's costs and benefits in resource-constrained scenarios.
>
> We thank the reviewer for this important point. We agree that our original efficiency discussion emphasises latency but does not separately report memory cost. ​​Since WASH uses multi-model ensembling, there is a clear **memory-latency trade-off**. To support different resource budgets, we provide two deployment variants:
>
> - **WASH-Parallel**: parallel multi-model decoding with synchronised KV caches, with $O(N)$ GPU memory cost and $O(1)$ per-token latency;
>
> - **WASH-Sequential**: sequential multi-model decoding on a single GPU with memory reuse, with $O(1)$ GPU memory cost and $O(N)$ per-token latency.
>
> We report the empirical results below with a blending size of 3 for long-text generation (512 tokens) using bf16 models:
>
> | Method Variant | Peak memory | Per-token Latency |
> |-|-:|-:|
> | Unwatermarked | 15.09 GB (1.00x) | 29.5 ms (0.98x) |
> | Watermarked     | 15.10 GB (1.00x) | 30.0 ms (1.00x) |
> | De-mark             | 16.58 GB (1.10×) | 190.6 ms (6.35×) |
> | ToBlend              | 71.46 GB (4.73×) | 172.3 ms (5.74×) |
> | WASH-Parallel   | 40.87 GB (2.71×) | 56.9 ms (1.90×) |
> | WASH-Sequential | 15.76 GB (1.04×) | 165.5 ms (5.52×) |
>
> These results confirm the trade-off discussed above. Also, under both deployment regimes, WASH remains more efficient than the prior removal baselines.

---

> > ### Author Rebuttal · Reviewer_SfU4 · 2026-04-04
> >
> > I thank the authors for the rebuttal, and I decide to keep my score .

---

> > > ### Author Response · Authors · 2026-04-06
> > >
> > > Thank you very much for your careful consideration of our rebuttal and for raising these important concerns. We appreciate your thoughtful questions, particularly those regarding robustness under defensive strategies and resource trade-offs.
> > >
> > > We hope that our clarifications on the framework’s robustness and its resource trade-off analysis provide a clearer understanding of the paper and further strengthen the work. We will incorporate these clarifications and additional results into the revised version to improve clarity.

---

### Official Review · Reviewer_axXy · 2026-03-11

**Soundness:** 3
**Presentation:** 3
**Significance:** 3
**Originality:** 3
**Overall Recommendation:** 4
**Confidence:** 3

**Summary:**

This paper studies the fragility of LLM watermarking under a realistic multi-provider setting. The key observation is that many watermarking methods can be viewed as inducing small perturbations to a model's output distribution; when an adversary can query multiple models in parallel, averaging (linear ensembling) of the output distributions can cancel independent perturbations and recover an approximately unwatermarked distribution. To address practical obstacles (e.g., tokenizer/vocabulary mismatch across heterogeneous models), the paper proposes WASH, which combines statistical hybridisation with fluency-aware routing and context re-synchronisation to enable efficient probability aggregation. Experiments report improved watermark attenuation and favorable quality/efficiency trade-offs.

**Compliance With Llm Reviewing Policy:**

Affirmed.

**Final Justification:**

The response partially resolved my issue, so I am maintaining my rating.

**Key Questions For Authors:**

See weaknesses!

**Limitations:**

Please add the limitations.

**Strengths And Weaknesses:**

Strengths:

1. The multi-provider access assumption is realistic in today's LLM ecosystem and is often under-emphasized in prior watermarking work.

2. The paper offers an intuitive, distributional view of watermarking and connects watermark attenuation under ensembling to concentration phenomena.

3. The tokenizer mismatch issue is real; proposing a method that attempts to make heterogeneous ensembling practical is valuable.

Weaknesses:

1. The core idea---attenuating watermark signals by averaging independent perturbations across multiple sources---appears conceptually close to techniques that have already been explored in the \emph{image} watermark removal literature (e.g., multi-source averaging/aggregation to suppress watermark-like components or independent artifacts).

2. The theoretical framing treats cross-provider watermark perturbations as i.i.d.\ and zero-mean. This assumption may be overly strong: 1) Many watermark schemes are not obviously unbiased; some intentionally skew token selection toward a greenlist or impose structured biases. 2) "Independence" may not hold if providers adopt correlated design choices, shared watermark families, or similar decoding constraints. Without such mapping, it is difficult to interpret whether the theory explains the empirical results or whether the results rely on additional unspoken properties.

3. The proposed attack relies on access to probability distributions (or logits) to perform averaging. In many real-world deployments, APIs only return text and do not expose token-level probabilities. Under a strict black-box setting, does WASH remain practical?

4. For white-box/open-source LLMs, an adversary can often fine-tune, distill, or otherwise modify the model, and may not need to "remove" a watermark at generation time. Conversely, if watermarking is applied in the decoding layer rather than model weights, an adversary with full access can disable it entirely. The paper should clarify the threat model and whether watermark insertion/removal remains a meaningful objective in such settings.

5. Table~2 suggests that WASH achieves generation quality comparable to (or even exceeding) the unwatermarked baseline. This raises the concern that WASH may be doing more than watermark attenuation---it may be implicitly altering outputs in a way that could introduce semantic drift. Currently, the paper does not provide clear human evaluation or explicit semantic alignment metrics to demonstrate that semantics are preserved while removing watermark signals.

6. Missing comparison to a strong, simple baseline: single-pass paraphrase/rewrite by an unwatermarked model.

---

> ### Author Rebuttal · Authors · 2026-03-31
>
> > **Q1**. Similarity to image watermark removal
>
>
> We agree they share some high-level intuition, and we will cite and discuss this more explicitly. The key difference is that image methods average in a shared pixel space, while LLMs do not share a common token space across providers. Our contributions are the LLM-specific formulation and a practical method for heterogeneous ensembling despite the aforementioned tokeniser mismatch.
>
> > **Q2**. Assumptions on unbiasedness and independence
>
> Please see our responses to **Reviewer SfU4, Q1 and Q2**. To summarise: we have extended the **theoretical** discussion beyond the independence and zero-mean assumptions. We also added **empirical** analysis of defensive strategies that intentionally violate these assumptions. These additions directly address the sensitivity of our results to weakened assumptions.
>
> > **Q3**. The proposed attack relies on access to probability distributions (or logits) to perform averaging.
>
> Please see our response to **Reviewer RueE, Q3**. To summarise: WASH is most efficient when next-token distributions are available, but it can also be extended to involve black-box for watermark removal. The limitation is that black-box models cannot participate as routing specialists unless their vocabulary is known, so the hybrid version loses some routing flexibility and efficiency.
>
> > **Q4**. Under white-box access, is generation-time watermark removal still meaningful (given fine-tuning/distillation or direct disabling)?
>
> We agree that if an attacker has full knowledge of the watermark mechanism and can directly modify the decoding layer, then disabling it is a strictly stronger attack. However, this is not the threat model considered here.
>
> Our focus is a more realistic inference-time adversary who can run/query the model for generation, but does **not** know the watermark type, key, detector internals, or even whether watermarking is present. In this setting, watermark removal remains meaningful, since the attacker cannot simply turn off the watermark or optimise directly against the detector. Compared to retraining-based attacks such as fine-tuning or distillation, WASH is training-free, does not modify model weights, and operates entirely at generation time.
>
> > **Q5**. Whether WASH preserves semantics
>
>
> We agree that watermark removal should not come at the cost of semantic drift. While our original paper evaluates task accuracy on reasoning and QA benchmarks, we further add **WritingBench** [1] to assess output quality in an open-ended generation setting. WritingBench uses LLM-as-a-judge scoring to evaluate generated responses with respect to the two semantic metrics, style and format. As shown in the table, WASH achieves superior generation quality in this open-ended task.
>
> | Method               | Style ↑ | Format ↑ | Per-token Latency ↓ |
> | - | -: | -: | -: |
> | Unwatermarked  | 4.02 |    4.13   | 0.92×   |
> | Watermarked      | 4.03 |    4.17   | 1.00×   |
> | De-mark              | 3.88 |    4.20   | 43.98× |
> | ToBlend               | 2.18 |    2.45   | 9.08×   |
> | Paraphrase         | 3.59 |    3.63   | 10.06× |
> | WASH                | 4.38 |    4.13   | 1.69×   |
>
> [1] WritingBench: A Comprehensive Benchmark for Generative Writing, NeurIPS 2025.
>
> > **Q6**. Missing comparison to a strong, simple baseline: single-pass paraphrase/rewrite by an unwatermarked model.
>
> We agree that single-pass rewriting with an equivalent unwatermarked model would be a strong baseline. However, access to such a model imposes significantly stronger attacker assumption and undermines the purpose of watermarking. In contrast, WASH operates without requiring such access.
>
> We thus compare against a strong paraphrasing baseline [2], which uses a weaker unwatermarked model to rewrite spans when a watermark is detected. This baseline assumes stronger access than WASH, as it requires both accurate online detection during the attack and access to an unwatermarked model.
>
> We evaluate on GSM8K and WritingBench to cover reasoning and creative writing:
>
> |Method|GSM8K Accuracy↑|WritingBench Score↑|Detect Score$^★$↓|
> |-|--:|--:|--:|
> |Unwatermarked|0.567|4.07|-0.02|
> |Watermarked|0.511|4.10|7.03|
> |De-mark|0.550|4.04|2.56|
> |ToBlend|0.568|2.32|1.44|
> |**In-Place Paraphrase**|0.467|3.61|1.83|
> |WASH|0.698|4.26|1.49|
>
> As shown above, paraphrasing reduces detectability but noticeably harms output quality, especially on GSM8K. WASH achieves a better overall performance on both quality and watermark removal.
>
> We also point to our response to **Reviewer RueE, Q2** for comparisons with other baselines.
>
> $^★$ The detector in [2] does not apply to paraphrasing. We therefore adopt the detector from [1]; see our response to Reviewer RueE, Q2 for details.
>
> [1] Zhang et al., Watermarks in the Sand: Impossibility of Strong Watermarking for Generative Models, ICML 2024.
>
> [2] Liu et al., Can Watermarked LLMs be Identified by Users via Crafted Prompts?, ICLR 2025.

---

> > ### Author Rebuttal · Reviewer_axXy · 2026-04-04
> >
> > Thank you for your response! The response partially resolved my issue, so I am maintaining my rating.

---

> > > ### Author Response · Authors · 2026-04-05
> > >
> > > Thank you for your continued engagement and for maintaining a positive assessment of our work. We sincerely appreciate your careful reading of the paper and the constructive questions you raised, which have helped us improve and strengthen the paper. We will ensure that all additional experiments and clarifications discussed during the rebuttal are incorporated into the revised version.

---

### Official Review · Reviewer_RueE · 2026-03-13

**Soundness:** 2
**Presentation:** 3
**Significance:** 2
**Originality:** 2
**Overall Recommendation:** 4
**Confidence:** 4

**Summary:**

The paper shows that distribution-based text watermarks can be removed by mixing outputs from several independently watermarked language models. It proves that the average of next-token distributions converges to the unwatermarked consensus distribution as the number of models increases. It also introduces WASH, a practical ensembling method that handles different tokenisers through routing and context re-synchronisation, and reports experiments across three models and six watermark schemes where watermark detection drops below thresholds with about three models while maintaining good quality and efficiency.

**Compliance With Llm Reviewing Policy:**

Affirmed.

**Final Justification:**

The paper presents WASH, a method for removing distribution-based text watermarks by ensembling next-token distributions from independently watermarked language models. The theoretical contribution, showing that such averaging converges to the unwatermarked distribution, is sound and well-supported by experiments across multiple watermark schemes.

My initial concerns were that the positioning of WASH as an attack was unclear and that comparisons against simpler baselines were missing. The authors addressed both in the rebuttal by explicitly framing WASH as an adaptive attack and by showing that naive averaging and paraphrasing variants perform substantially worse. This resolved my main objections and led me to raise my score to “Weak Accept”.

**Key Questions For Authors:**

1. **Is WASH intended as an attack model for watermark removal, or as a practical generation strategy that users would actually adopt?**
   Please clarify this explicitly in the introduction and threat model, since this affects how the contribution should be interpreted.

2. **How much of the reported benefit comes from the full WASH design, rather than from simpler attacks?**
   In particular, can you compare against naive averaging and naive averaging followed by paraphrasing, both in watermark removal and in text quality?

3. **What exact access does the attacker need to run WASH?**
   Does the method require next-token probability distributions from all models, or can it work with only sampled outputs? This would significantly affect the practicality of the attack.

**Limitations:**

Partially. The paper includes an impact statement, but it does not fully discuss misuse risk. The method can be used to remove watermarks and weaken provenance and attribution tools. The paper should state this risk clearly and describe realistic attacker capabilities and access assumptions.

**Strengths And Weaknesses:**

**Strengths**. The paper gives a clear idea. Watermarks change the distribution of the next token, and averaging across independent models can cancel these changes. The theory aligns with this idea and explains why the signal drops as more models are added. The experiments cover several watermark schemes and several base models, and they show large drops in detection scores. The method also tries to handle different tokenisers with a concrete routing and resynchronisation procedure, and the paper reports quality and speed results in addition to detection.

**Weaknesses**. The main weakness is the positioning of WASH. The paper presents WASH as a practical method for removing watermarks while preserving fidelity, but it is not clear whether this should be viewed as a realistic user workflow or as an explicit attack. In normal use, people do not usually average next-token distributions across several LLMs, so the introduction should clarify this setting and specify the attacker's required capabilities. The empirical evaluation is also incomplete in this regard. Since the core idea is averaging across models, the paper should compare WASH against simpler baselines such as naive averaging and naive averaging followed by paraphrasing, to show whether the added routing and re-synchronisation machinery is truly necessary. The paper also relies on strong assumptions, especially independence and zero-mean perturbations across providers, and it does not examine how sensitive the results are when these assumptions are weakened.

---

> ### Author Rebuttal · Authors · 2026-03-31
>
> > **Q1**. Is WASH intended as an attack model for watermark removal, or as a practical generation strategy that users would actually adopt?
>
> Thank you for raising this important distinction. WASH is intended as an adaptive watermark-removal attack model in a multi-provider setting, rather than as a generation strategy that ordinary users would naturally adopt in everyday use.
>
> Our threat model assumes a motivated attacker who can query multiple LLMs and ensemble their next-token distributions to suppress provider-specific watermark perturbations. In this setting, additional inference cost is acceptable to the extent that it materially improves removal effectiveness. Relative to prior watermark-removal attacks, which are often query-intensive or computationally prohibitive for long-form generation, WASH makes this stronger attack substantially more practical while preserving text quality. We will clarify this positioning more explicitly in the revision.
>
> > **Q2**. Can you compare against naive averaging and naive averaging followed by paraphrasing, both in watermark removal and in text quality?
>
>
> Thanks for the suggestion. We add two baselines: **naive token-level averaging** and **averaging + paraphrasing** , with paraphrasing applied using both watermarked and unwatermarked models. We evaluate on GSM8K and WritingBench to cover reasoning and creative writing:
>
> |Method|GSM8K Accuracy↑|WritingBench Score↑|Detect Score$^★$↓|
> |-|-:|-:|-:|
> |Unwatermarked|0.567|4.07|-0.02|
> |Watermarked|0.511|4.10|7.03|
> |De-mark|0.550|4.04|2.56|
> |ToBlend|0.568|2.32|1.44|
> |**Average**|0.339|3.95|1.53|
> |**Average + WM Paraphrase**|0.197|4.02|2.70|
> |**Average + Un-WM Paraphrase**|0.205|4.05|0.26|
> |WASH|0.698|4.26|1.49|
>
> As shown above, naive averaging performs poorly on both tasks, as it does not properly re-synchronise model outputs and thus leads to inconsistent reasoning. Neither paraphrase variants can resolve this issue and exhibits even worse GSM8K accuracy.
>
> We note the fundamental issues of both watermarked and unwatermarked paraphrasing. For the former, it simply reintroduces watermarked signals. For the latter, the effective removal of watermarks is possible thanks to white-box access to an equally strong unwatermarked model. This undermines the purpose of attacking. In contrast, WASH operates without requiring such access.
>
> Overall, WASH achieves the best trade-off between watermark removal.
>
> $^★$ Watermark detector used in paper [2] ensures robustness and generalisability by identifying the model’s logit perturbation on a small set of generated tokens, but it does not hold under paraphrasing due to the reliance on longer spans. We thus use the detector adopted in [1].
>
> [1] Zhang et al., Watermarks in the Sand: Impossibility of Strong Watermarking for Generative Models, ICML 2024.
>
> [2] Liu et al., Can Watermarked LLMs be Identified by Users via Crafted Prompts?, ICLR 2025.
>
> > **Q3**. What exact access does the attacker need to run WASH? Does the method require next-token probability distributions from all models, or can it work with only sampled outputs?
>
> We thank the reviewer for this important question. In its most efficient form, WASH assumes access to next-token distributions, since this enables direct distribution ensembling and the efficiency optimisations described in the paper.
>
> That said, WASH can indeed be extended to a hybrid white-box/black-box regime, as we illustrate next. Suppose we have $X$ white-box models and $N-X$ black-box models. During standard ensemble decoding, we can sample from the white-box group with probability $X/N$ using the distribution ensemble, and otherwise uniformly select one black-box model to generate the next token. The sampled token is then fed back to all models for context alignment. This is distributionally equivalent to averaging over all $N$ models at the token level.
>
> However, during fluency-aware routing, black-box models cannot serve as specialist candidates unless their vocabulary is known or exposed, since routing requires checking token compatibility across models. This makes the black-box extension a restricted version of WASH: it remains compatible with watermark removal, but loses some routing flexibility and efficiency. We will clarify this limitation in the revision.
>
> > **Q4**. The paper also relies on strong assumptions, especially independence and zero-mean perturbations across providers, and it does not examine how sensitive the results are when these assumptions are weakened.
>
> Due to space constraints, we only give the main points here. We have extended the **theoretical** discussion beyond the independence and zero-mean assumptions (see **Reviewer SfU4, Q1**), and we also added **empirical** analysis of defensive strategies that intentionally violate these assumptions (see **Reviewer SfU4, Q2**). These additions directly address the sensitivity of our results to weakened assumptions.

---

> > ### Author Rebuttal · Reviewer_RueE · 2026-04-04
> >
> > I appreciate the responses provided. The authors' positioning of WASH as an explicit attack and the subsequent comparison against naive averaging and alternative watermark removal techniques significantly strengthen the paper's contribution. Given this improved understanding of the work, I will be updating my score.

---

> > > ### Author Response · Authors · 2026-04-05
> > >
> > > Thank you very much for your thoughtful review, which has genuinely helped us strengthen the contribution of our paper during the rebuttal. We will further clarify the positioning of the paper and incorporate the additional experiments comparing with baselines in the revised version. We are glad that our responses have helped address your concerns, and we would be very grateful if you would update your score to reflect your latest assessment before the discussion period ends.

---

### Decision · Program_Chairs · 2026-04-30

**Decision:**

Accept (regular)

**Comment:**

The paper shows that distribution-based text watermarks can be removed by mixing outputs from several independently watermarked language models. It proves that the average of next-token distributions converges to the unwatermarked consensus distribution as the number of models increases. It also introduces WASH, a practical ensembling method that handles different tokenisers through routing and context re-synchronisation, and reports experiments across three models and six watermark schemes where watermark detection drops below thresholds with about three models while maintaining good quality and efficiency.

Reviewers like the following strengths:
- The paper introduces a novel method to remove watermarks across six different SOTA watermarking schemes.
- The paper provides a theoretical justification for the removal of watermarks under certain assumptions.

Reviewers also raised concerns about:
- The problem setup and motivation. In reality, users do not average the token probabilities from multiple LLMs (since they are not usually provided).
- Too strong assumptions, especially independence and zero-mean perturbations across providers, and it does not examine how sensitive the results are when these assumptions are weakened. Different watermark providers will apply different watermarking schemes and keys.